# Transient PU.1 low fetal progenitors generate lymphoid progeny that contribute to adult immunity

Encarnacion Montecino-Rodriguez, Oscar I Estrada, Kenneth Dorshkind

**Hematopoietic stem cells and multipotential progenitors emerge in multiple, overlapping waves of fetal development. Some of these populations seed the bone marrow and sustain adult B- and T-cell development long-term after birth. However, others are present transiently, but whether they are vestigial or generate B and T cells that contribute to the adult immune system is not well understood. We now report that transient fetal progenitors distinguished by expression of low levels of the PU.1 transcription factor generated activated and memory T and B cells that colonized and were maintained in secondary lymphoid tissues. These included the small and large intestines, where they may contribute to the maintenance of gut homeostasis through at least middle age. At least some of the activated/memory cells may have been the progeny of B-1 and marginal zone B cells, as transient PU.1^{low} fetal progenitors efficiently generated those populations. Taken together, our data demonstrate the potential of B- and T-cell progeny of transient PU.1^{low} fetal progenitors to make an early and long-term contribution to the adult immune system.**

## Introduction

Lymphocyte development initiates in utero and continues after birth in the bone marrow and thymus (Montecino-Rodriguez & Dorshkind, 2012; Kobayashi & Yoshimoto, 2023), but lymphopoiesis is not a linear process. Instead, hematopoietic stem cells (HSCs), multipotential progenitors, and more restricted progenitors with lymphoid potential emerge in multiple, overlapping waves of development (Patel et al, 2022; Kobayashi et al, 2023). Some of these stem and progenitor cells seed the bone marrow and sustain adult B- and T-cell development (Patel et al, 2022; Kobayashi et al, 2023; Kobayashi & Yoshimoto, 2023). However, others are present only transiently in utero or for a short period in the neonate and are not maintained long-term (Montecino-Rodriguez et al, 2006; Beaudin et al, 2016).

We previously showed that some of the transient fetal lymphoid progenitors were distinguished by low expression of *Spi1*, the gene

that encodes the PU.1 transcription factor (Montecino-Rodriguez et al, 2016, 2018). In this regard, PU.1^{low} fetal progenitors were present in WT mice as well as in a strain of mice (Rosenbauer et al, 2006) engineered to express low levels of *Spi1* by deletion of an upstream regulatory element (URE) located 14 kb from the *Spi1* transcription start site (Montecino-Rodriguez et al, 2016). However, they were not maintained long-term after birth in the bone marrow of these so-called UREΔ/Δ mice. In addition to showing that PU.1^{low} fetal progenitors were present only transiently, our analyses also demonstrated that in agreement with the initial report describing this strain (Rosenbauer et al, 2006), adult lymphocyte development was blocked in UREΔ/Δ mice (Montecino-Rodriguez et al, 2016, 2018), thus confirming the known dependence of that process on the PU.1 transcription factor (Pang et al, 2018).

Whether transient PU.1^{low} fetal progenitors are vestigial or produce B and T cells that contribute to the adult immune system is not well understood. There is precedent for considering that they may do so because transient yolk sac progenitors that emerge between E8.5 and E16.5 are the source of adult tissue macrophages (Schulz et al, 2012; Gomez Perdiguero et al, 2015). Interestingly, and by analogy to the differences in levels of *Spi1* expression between fetal and adult lymphoid progenitors, macrophage development from transient fetal myeloid progenitors is not dependent on the Myb transcription factor but is necessary for adult monocyte/macrophage production (Schulz et al, 2012). We were thus interested to determine if transient PU1^{low} fetal progenitors generated B and T cells that contributed to the adult immune system. The UREΔ/Δ strain is an ideal model system to do so because the latter populations can be tracked in the absence of adult lymphocyte development.

Our initial analyses of UREΔ/Δ mice showing that transient PU.1^{low} fetal progenitors had lymphoid potential were based primarily on in vitro analyses (Montecino-Rodriguez et al, 2016, 2018). We, therefore, examined UREΔ/Δ mice through middle age to determine the contribution of transient PU.1^{low} fetal progenitors to the adult immune system in vivo. We now report that they generated activated and memory T and B cells that were maintained in secondary lymphoid tissues through at least middle age and that a significant proportion of the latter cells were likely the progeny of innate B-1 and marginal zone (MZ) B cells. In addition, the data

Department of Pathology and Laboratory Medicine, David Geffen School of Medicine at UCLA, Los Angeles, CA, USA

Correspondence: kdorshki@mednet.ucla.edu

showed that the B- and T-cell progeny of PU.1[low] progenitors colonized the intestines of UREΔ/Δ mice and, even in the absence of B and T cells generated from definitive adult HSCs and multipotential progenitors, they had the potential to maintain gut homeostasis through middle age.

Taken together, our data demonstrated that PU.1[low] fetal progenitors that arise in transient waves of fetal lymphopoiesis generated B- and T-cell progeny that made an early and long-term contribution to the adult immune system. The data further suggested that, similar to what has been observed for macrophages, the B and T cells present in adult tissues are the progeny of transient fetal and adult stem and progenitor cells whose development is regulated by distinct transcriptional programs.

## Results

### PU.1[low] fetal progenitors generate activated and memory B and T cells through middle age

In contrast to mice with a germ-line deletion of *Spi1* that die in utero or soon after birth (Scott et al, 1994; McKercher et al, 1996), UREΔ/Δ mice have no overt pathology despite the fact that lymphocyte development from the adult stem and progenitor cells is blocked because of an 80% reduction in expression of *Spi1* (Rosenbauer et al, 2006; Montecino-Rodriguez et al, 2016, 2018). Further evidence of this developmental block was provided by analysis of early B- and T-cell progenitors in the bone marrow (BM) and thymus of UREΔ/Δ mice (Figs S1 and S2).

We performed FACS analysis of B-cell progenitors in the marrow based on the scheme developed by Hardy et al (1991) (Fig S1A). This analysis indicated that few CD45R(B220)$^+$CD43$^+$CD19$^-$CD24$^-$Ly51$^-$ Fraction A pre-pro-B cells were present in 4-wk–old UREΔ/Δ bone marrow. CD45R(B220)$^+$CD43$^+$CD19$^+$CD24$^+$Ly51$^{-/+}$ Fraction B and C-C′ pro-B cells and CD45R(B220)$^+$CD43$^-$ pre-B cells were present in the marrow from 4-wk–old UREΔ/Δ mice, but their frequency was lower compared with WT animals. The fact that some B-cell progenitors were present in UREΔ/Δ mice 4 wk after birth is consistent with the known survival of some transient fetal derived stem and progenitor cells through young adulthood (Montecino-Rodriguez et al, 2006; Bowie et al, 2007). However, by middle-age B-cell development was non-existent in UREΔ/Δ mice because no pre-pro-B, pro-B, or pre-B cells were detected (Fig S1B and C).

A similar pattern was observed for T-cell development (Fig S2). Cell counts revealed that a major loss of thymocytes was already evident in 4–6-wk–old UREΔ/Δ mice, and almost no cells were present in 11-mo–old animals (Fig S2A). Based on the FACS gating strategy shown in Fig S2B, we observed that the frequency of lineage negative (Lin)$^-$CD117$^+$CD44$^+$CD25$^-$ early T-lineage progenitors (ETPs) in the thymus from 4-wk–old UREΔ/Δ was eightfold lower than in WT mice, and by middle age ETPs were not detected (Fig S2C). The sharp decline in the number of thymocytes in UREΔ/Δ mice by 4 wk of age and the severe loss of T-cell progenitors is distinct from the pattern of decreased thymopoiesis that typifies thymic involution. For example, we detected ETPs in 50-wk–old WT mice, consistent with a previous report from our laboratory (Min et al, 2004), but they as well as Lin$^-$CD44$^+$CD25$^+$

double negative (DN) 2 and Lin$^-$CD44$^-$CD25$^+$ DN3 populations were totally absent from UREΔ/Δ mice by 24 wk of age (Fig S2C).

Nevertheless, the normal health span of UREΔ/Δ mice suggested that immune cells generated from transient PU.1[low] fetal progenitors must have provided some degree of immune protection. We initially performed a global analysis and quantified B and T cells in the spleen (SPL) of middle-aged (22–46 wk old) UREΔ/Δ mice to assess this possibility. The analysis of B-cell subpopulations in that organ using gating strategies shown in Fig S3A and B revealed that CD93$^-$ CD23$^+$ CD21$^{low}$ surface (s)IgM$^+$ naïve follicular (FO) B cells were present, but their frequency and total number were significantly lower than in middle-aged WT mice (Fig 1A and B). Instead, most of the cells in the SPL of UREΔ/Δ mice included plasma cells/plasmablasts (PC/PBs), class-switched cytoplasmic (c) IgA$^+$ cells, and CD80$^+$ activated/memory B cells; the latter two populations were present at WT levels (Figs 1A and B and S3C).

The level at which CD44 (Baaten et al, 2010; Schumann et al, 2015) and CD62L (Anderson et al, 2003) are expressed has been used to identify CD62L$^{hi}$CD44$^{hi}$ central memory (T$_{CM}$), CD62L$^{low}$CD44$^{hi}$ effector memory (T$_{EM}$) (Benichou et al, 2017), CD62$^{low}$CD44$^{low}$ pre-effector memory (T$_{pEM}$) (Nakajima et al, 2021), and CD62L$^{hi}$CD44$^{-/low}$ naïve (T$_N$) CD4$^+$ and CD8$^+$ T cells. We quantified these T-cell subsets using gating strategies shown in Fig S4A in the SPL of middle-aged UREΔ/Δ mice and showed that the total number of CD4$^+$ and CD8$^+$ T$_N$ cells was significantly lower than in similarly aged WT mice. However, CD4$^+$ and/or CD8$^+$ T$_{CM}$, T$_{EM}$, and T$_{pEM}$ cells were present, often at WT levels (Figs 1C and D and S4B).

### PU.1[low] fetal progenitors generate an early, transient wave of naïve B and T cells

The above results showing that only low numbers of naïve FO B and CD4$^+$ and CD8$^+$ T$_N$ cells were present in the SPL of middle-aged UREΔ/Δ mice are consistent with the fact that PU.1[low] progenitors are not maintained long-term after birth and that lymphoid development from adult stem and progenitor cells is blocked in that strain (Figs S1 and S2). To determine if PU.1[low] fetal progenitors had the potential to generate naïve T and B cells during early life, lymphoid tissues from young (4–20 wk old) UREΔ/Δ mice were examined.

The number of cells in the SPL of young UREΔ/Δ mice was significantly higher than in middle-aged mice, although the total cell number was lower than in WT animals regardless of age (Fig 2A). As in middle-aged mice, the number of CD93$^-$ CD23$^+$ CD21$^{low}$ sIgM$^+$ FO B cells was significantly lower than in WT mice (Fig 2B and C). However, the number of FO B cells was significantly higher ($P = 0.0002$) in young compared with middle-aged UREΔ/Δ mice (compare Figs 1B and 2C). PC/PBs, class-switched cIgA$^+$ cells, and CD80$^+$ activated/memory B cells were also present in the SPL of young UREΔ/Δ mice (Fig 2B and C).

We also detected naïve CD4$^+$ and CD8$^+$ T cells in the SPL of young UREΔ/Δ mice (Fig 2D and E). The number of CD4$^+$ T$_N$ cells ($P < 0.001$) and CD8$^+$ T$_N$ cells ($P < 0.0001$) was significantly higher in young compared with middle-aged animals (compare Figs 1D and 2E).

Taken together, these results indicated that PU.1[low] fetal progenitors produced naïve B and T cells that colonized young UREΔ/Δ SPL. However, their production was not sustained, as PU.1[low] fetal progenitors were present only transiently, and by middle age numbers of naïve B and T cells had waned. We predicted that this

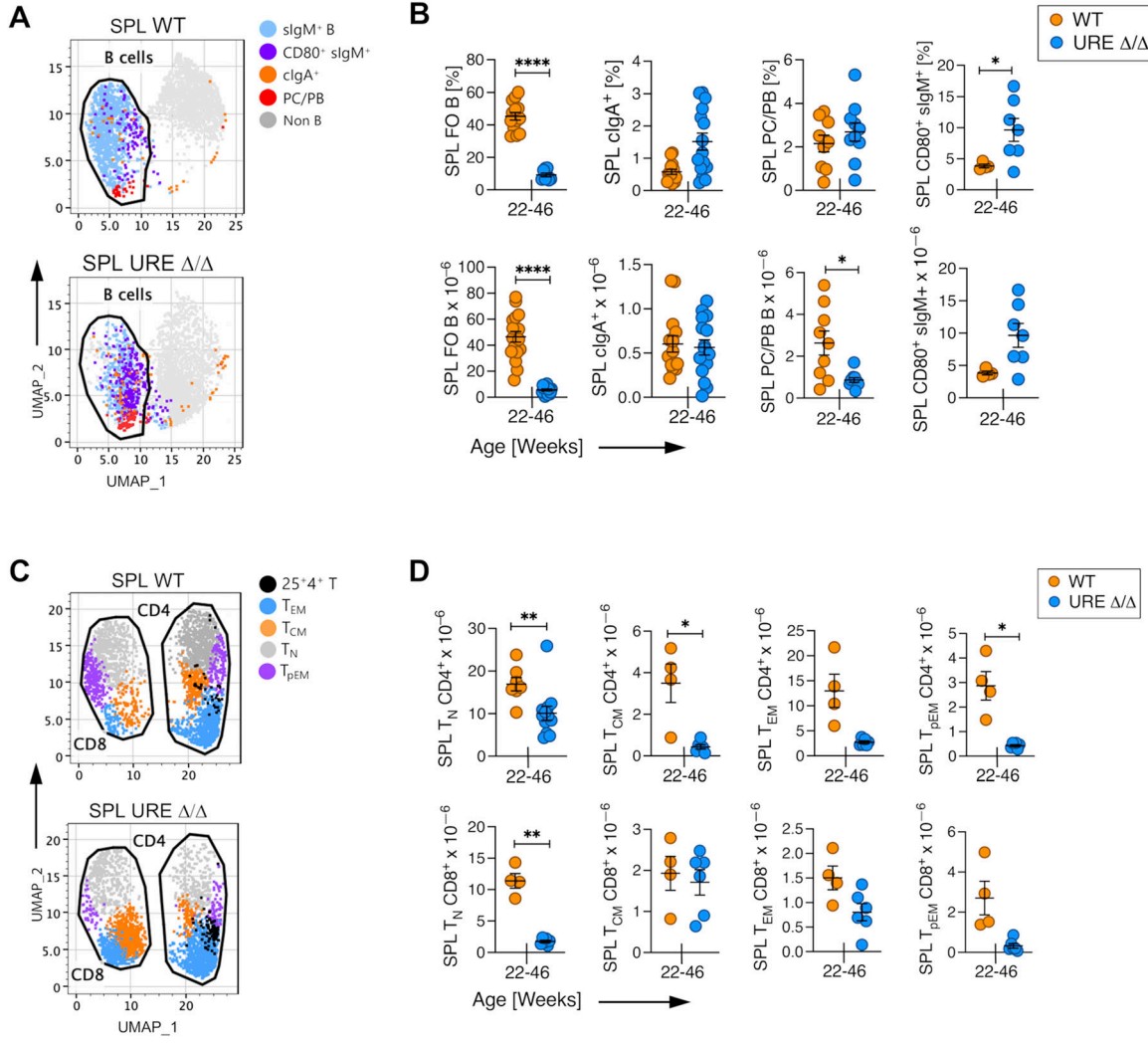

**Figure 1.** **PU.1$^{low}$ progenitors generate activated and memory B and T cells through middle age.**
**(A)** Uniform Manifold Approximation and Projection visualization of FACS data of B lineage cells in the spleen of middle-aged UREΔ/Δ and WT mice. **(B)** Frequency and number of FO IgM$^+$ and cIgA$^+$ B cells and PB/PC and CD80$^+$ B cells in the SPL of middle-aged UREΔ/Δ and WT mice. **(C)** Uniform Manifold Approximation and Projection visualization of FACS data of CD62L$^{hi}$CD44$^{low}$ naïve (T$_N$), CD62L$^{hi}$CD44$^{hi}$ central (T$_{CM}$), CD62L$^{low}$CD44$^{hi}$ effector (T$_{EM}$) memory, and CD62L$^{low}$CD44$^{low}$ pre-effector memory (T$_{pEM}$) T-cell subpopulations in the SPL of middle-aged UREΔ/Δ and WT mice. **(D)** Numbers of T$_N$, T$_{CM}$, T$_{EM}$, and T$_{pEM}$ CD4$^+$ and CD8$^+$ T cells in the SPL of middle-aged UREΔ/Δ and WT mice. Each circle represents an individual mouse. *$P$ < 0.05; ****$P$ < 0.0001 ($t$ test).

would have a major impact on the maintenance of mesenteric lymph nodes (MLNs) and Peyer's patches (PPs). Their initial organogenesis requires lymphoid tissue inducer and initiator cells, but once established, naïve B and T cells assume an important role in maintaining the differentiation and survival of mesenchymal and stromal cells that form their infrastructure (Randall et al, 2008).

MLNs were present in young animals, although, as in the SPL, the number of cells present was significantly lower than in WT mice (Fig 3A) and included FO and cIgA$^+$ B cells and PC/PBs; the latter cells were present at a significantly lower level than in WT controls (Fig 3B and C). Low numbers of CD4$^+$ and CD8$^+$ T$_N$, T$_{CM}$, T$_{EM}$, and T$_{pEM}$ cells were also detected in MLNs from young UREΔ/Δ mice but at significantly lower levels than in WT controls (Fig 3D and E). However, MLNs were difficult to identify in middle-aged UREΔ/Δ mice and were largely acellular (Fig 3A), and the limited number of cells that could be obtained precluded analysis.

PPs were more severely impacted than MLNs. Only a few PPs were identified in young UREΔ/Δ mice, and none were found in middle-aged animals, even by microscopic examination of the small intestine. Cell counts of PPs from young UREΔ/Δ mice indicated that they contained few cells compared with young WT tissues (Fig 3F). The limited analyses that could be performed indicated that the PPs from young UREΔ/Δ mice included sIgM$^+$ and cIgA$^+$ B cells and PB/PCs (Fig 3G and H) as well as CD4$^+$ and CD8$^+$ T, T$_{EM}$, and T$_{pEM}$ cells (Fig 3I and J).

## PU.1$^{low}$ fetal progenitors generate innate B cells that are maintained through middle age

We next assessed whether transient PU.1$^{low}$ fetal progenitors generated innate-like effectors. In view of our previous results indicating that they had marginal zone (MZ), B-1a, and B-1b B-cell

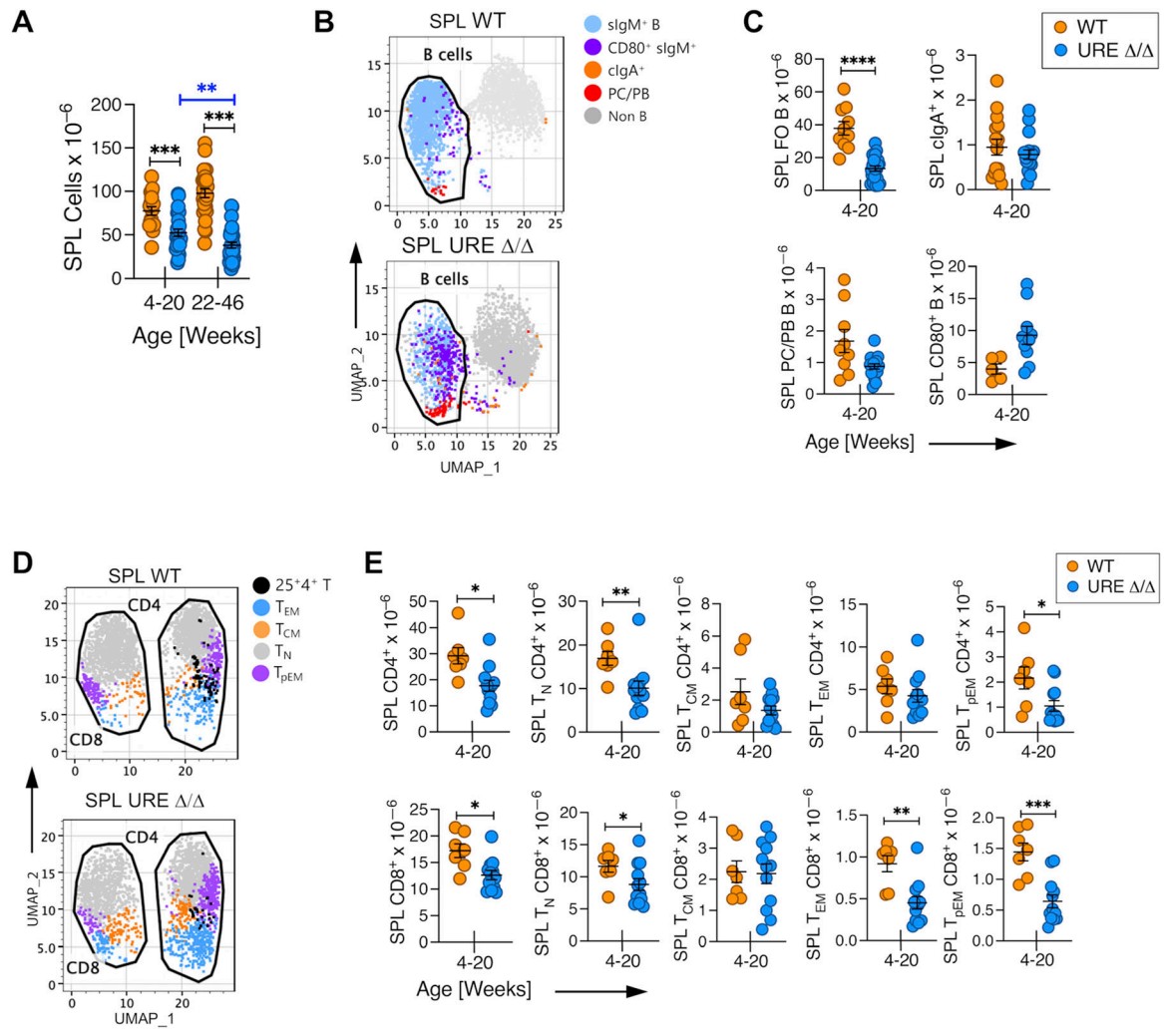

**Figure 2. PU.1^low progenitors generate an early, transient wave of naive B and T cells in the SPL.**

**(A)** Total number of cells harvested from SPL from young adult (4–20 wk old) and middle-aged (22–46 wk old) UREΔ/Δ and WT mice. **(B)** Uniform Manifold Approximation and Projection visualization of FACS data of B lineage cells in the SPL of young UREΔ/Δ and WT mice. **(C)** Frequency and/or number of FO IgM⁺ and cIgA⁺ B cells, and CD80 B cells and PB/PC in SPL of young UREΔ/Δ and WT mice. **(D)** Uniform Manifold Approximation and Projection visualization of FACS data of CD62L$^{hi}$CD44$^{low}$ naïve ($T_N$), CD62L$^{hi}$CD44$^{hi}$ central ($T_{CM}$), CD62L$^{low}$CD44$^{hi}$ effector ($T_{EM}$) memory, and CD62$^{low}$CD44$^{low}$ pre-effector memory ($T_{pEM}$) T-cell populations in the SPL of young UREΔ/Δ and WT mice. **(E)** Numbers of CD4⁺ and CD8⁺ T cells and $T_N$, $T_{CM}$, $T_{EM}$, and $T_{pEM}$ CD4⁺ and CD8⁺ subpopulations in SPL of young UREΔ/Δ and WT mice. Each circle represents an individual mouse. *$P < 0.05$; **$P < 0.01$; ***$P < 0.001$; ****$P < 0.0001$ ($t$ test).

potential (Montecino-Rodriguez et al, 2016), we focused on those populations.

B-1 B cells, and B-1a B cells in particular, can be generated from fetal progenitors (Kantor & Herzenberg, 1993; Montecino-Rodriguez & Dorshkind, 2012; Kobayashi et al, 2023). However, whether fetal derived B-1 B cells make a contribution to the adult B-1 B-cell compartment has been challenged by the identification of cells with B-1 potential in adult bone marrow (Berland & Wortis, 2002; Düber et al, 2009; Esplin et al, 2009), and a recent study concluding that postnatal rather than fetal lymphopoiesis is the predominant source of B-1a B cells in adult mice (Vergani et al, 2022). B-1a and B-1b B cells primarily localize to serous cavities such as the peritoneal cavity (PerC), and we used expression of CD5, CD23, and CD21 to identify them in that tissue (Fig 4A).

Both CD5⁺CD23⁻CD21$^{-/low}$ sIgM⁺ B-1a and CD5⁻CD23⁻CD21$^{-/low}$ sIgM⁺ B-1b B cells were present in the PerC of young and middle-aged UREΔ/Δ mice, and the total number of B-1 B cells as well as B-1a and B-1b subsets was similar and/or elevated, often significantly so, when compared with age matched WT animals (Fig 4B–D). B-1 B cells were also present in SPL of young UREΔ/Δ mice (Fig 4E). However, in contrast to what was observed in the PerC, by middle age their number was significantly reduced compared with middle-aged WT mice (Fig 4F).

We previously showed that CD93⁻ CD23⁻ CD21$^{hi}$ sIgM⁺ marginal zone (MZ) B cells (Fig 4E) were present in the SPL of young adult UREΔ/Δ mice (Montecino-Rodriguez et al, 2016) but did not determine if they were maintained long-term. We found that the total number of MZ B cells was similar in the SPL of UREΔ/Δ and WT mice regardless of age (Fig 4G). In fact, the frequency of MZ B cells was

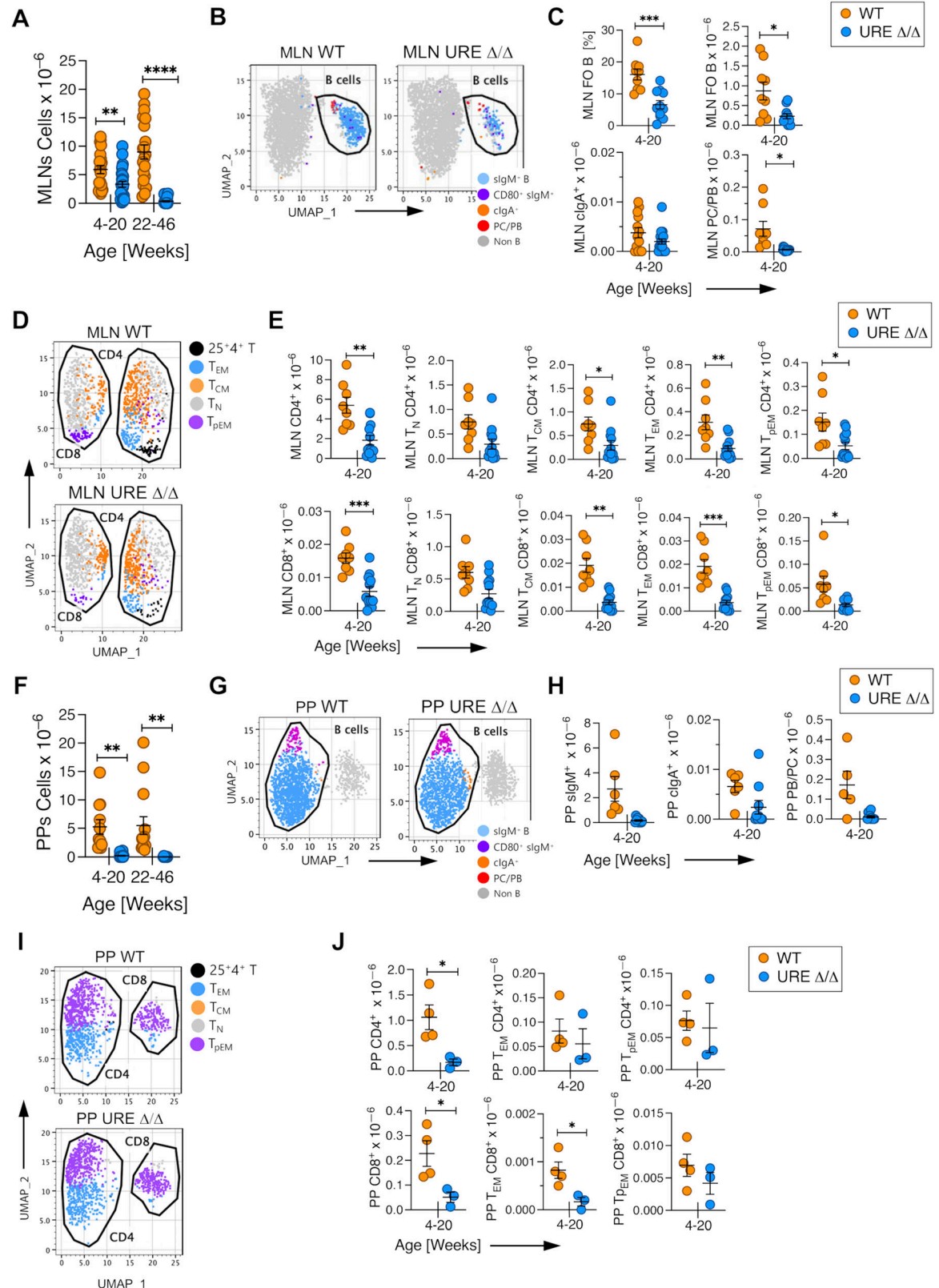

**Figure 3. PU.1^low progenitors generate B and T cells that colonize mesenteric lymph nodes (MLNs) and PPs.**
**(A)** Total number of cells harvested from MLNs from young adult (4–20 wk old) and middle-aged (22–46 wk old) UREΔ/Δ and WT mice. **(B)** Uniform Manifold Approximation and Projection (UMAP) visualization of FACS data of B lineage cells in MLNs from young UREΔ/Δ and WT mice. **(C)** Numbers of FO IgM^+ and cIgA^+ B cells, and

higher in UREΔ/Δ compared with WT mice regardless of age, which we attribute to the low number of naïve B-2 B cells present in UREΔ/Δ SPL (Fig 1B).

### Transient PU.1[low] fetal progenitors generated B and T cells that colonize the intestines

The intestines are colonized in utero by diverse populations of B and T cells (Spencer et al, 1986; Golby et al, 2002; Schreurs et al, 2019; Stras et al, 2019), including memory CD4[+] T cells (Li et al, 2019), but their origin has not been defined. We thus considered the possibility that transient PU.1[low] fetal progenitors were a source.

The analysis of young UREΔ/Δ mice indicated that cells harvested from the lamina propria of the small intestine (SI) included sIgM[+] and cIgA[+] B cells and PC/PBs that were present at WT levels (Fig 5A and B). WT levels of CD4[+] and CD8[+] T cells were also present and included CD4[+] and CD8[+] T$_{EM}$ and T$_{pEM}$ cells (Fig 5C and D). Similarly, WT levels of sIgM[+] and cIgA[+] B cells and PC/PBs (Fig 5E and F) and CD4[+] and CD8[+] T cells, including T$_{EM}$ and T$_{pEM}$ cells (Fig 5G and H), were present in the UREΔ/Δ large intestine (LI). A population of fetal-derived CD8[+] T cells with innate-like functions has been described and is distinguished by their CD44[high] CD122[high] phenotype (Smith et al, 2018). These cells may be present in the SI and LI of young UREΔ/Δ mice because some of the CD8[+] cells in these tissues expressed high levels of CD44 (data not shown).

Analysis of the intestines from middle-aged UREΔ/Δ mice indicated that these B- and T-cell populations were maintained, but often at significantly lower levels than in WT mice. For example, sIgM[+] B cells and PC/PBs cells were detected in the UREΔ/Δ SI, but their total number was significantly lower than in WT mice. However, the number of cIgA[+] cells was equivalent between UREΔ/Δ and WT mice (Fig 6A and B). Nevertheless, the number of cIgA[+] cells declined between young and middle age in UREΔ/Δ mice (compare Figs 5B and 6B). The total number of CD4[+] and CD8[+] T cells was also significantly lower in the SI from middle-aged UREΔ/Δ compared with WT mice, although T$_{EM}$ and T$_{pEM}$ cells were still present (Fig 6C and D).

The number of sIgM[+] B cells was also significantly lower in the LI of middle-aged UREΔ/Δ compared with WT mice, but WT levels of PC/PBs and cIgA[+] cells were present (Fig 6E and F). CD4[+] and CD8[+] T cells as well as T$_{EM}$ and T$_{pEM}$ cells were also found in the LI of middle-aged UREΔ/Δ mice, albeit at lower levels compared with WT mice, and CD4[+] and CD8[+] T$_{EM}$ and T$_{pEM}$ cells were present (Fig 6G and H).

Although PC/PBs were present in the intestines, they were not uniformly distributed in all tissues of UREΔ/Δ mice. In this regard, the bone marrow is a reservoir of plasma cells (Tellier et al, 2024), but no PCs were present in that tissue from UREΔ/Δ mice (Fig 6I).

That observation suggests that the presence of PC/PBs in the SI and LI was not an artifact because of a lack of competition from adult plasma cells. Instead, the data indicate that the PB/PCs derived from UREΔ/Δ stem/progenitor cells have distinct tissue-homing properties.

### The lymphoid progeny of transient PU.1[low] fetal progenitors may contribute to gut homeostasis

Taken together, the above results demonstrated transient PU.1[low] fetal progenitors were the source of various activated and memory B and T cells that colonized the intestines and were maintained through at least middle age. We next determined if these cells, in the absence of input of B and T cells generated from adult stem and progenitor cells, had the potential to maintain intestinal homeostasis. 16S rRNA sequencing of fecal samples from young and middle-aged WT and UREΔ/Δ mice housed in the same colony and fed identical diets was performed.

Subtle differences in the gut microbiome were observed between middle-aged UREΔ/Δ and WT mice, as shown in the relative abundance (Fig 7A) and the Shannon index (Fig 7B) plots. The primary difference between UREΔ/Δ and WT mice was the level at which some bacterial genera were present (Fig 7A). The relatively normal gut microbiome in UREΔ/Δ mice is likely because of the fact that class-switched IgA[+] B cells were present at almost WT levels in the Si and Li of middle-aged UREΔ/Δ mice (Fig 6B and F). The observation that young and middle-aged UREΔ/Δ and WT mice had similar concentrations of secreted IgA in their fecal samples (Fig 7C) is consistent with this conclusion.

## Discussion

Some fetal lymphoid progenitors express lower levels of *Spi1* than their adult counterparts, and these PU.1[low] populations are present transiently in UREΔ/Δ mice (Montecino-Rodriguez et al, 2016). The present study analyzed that strain to determine if these transient PU.1[low] fetal progenitor cells generated B- and T-cell progeny that contributed to the adult immune system. One advantage of this model is that lymphocyte development can be assessed in unmanipulated mice, thereby avoiding artifacts associated with the transplantation of restricted pools of fetal stem and progenitors into adult recipients preconditioned with drugs and/or irradiation. Whereas the initial report that described UREΔ/Δ mice (Rosenbauer et al, 2006), and subsequent studies from our laboratory (Montecino-Rodriguez et al, 2016, 2018) indicated that lymphopoiesis in the bone marrow and thymus of UREΔ/Δ mice was

---

PB/PCs in MLNs from young UREΔ/Δ and WT mice. **(D)** UMAP visualization of FACS data of CD62L[hi]CD44[low] naïve (T$_N$), CD62L[hi]CD44[hi] central (T$_{CM}$), CD62L[low]CD44[hi] effector (T$_{EM}$) memory, and CD62[low]CD44[low] pre-effector memory (T$_{pEM}$) T-cell populations in MLNs from young UREΔ/Δ and WT mice. **(E)** Numbers of CD4[+] and CD8[+] T cells and T$_N$, T$_{CM}$, T$_{EM}$, and T$_{pEM}$ CD4[+] and CD8[+] subpopulations in MLNs from young UREΔ/Δ and WT mice. **(F)** Total number of cells harvested from PPs from young adult (4–20 wk old) and middle-aged (22–46 wk old) UREΔ/Δ and WT mice. **(G)** UMAP visualization of FACS data of B lineage cells in PPs from young UREΔ/Δ and WT mice. **(H)** Number of IgM[+] and cIgA[+] B cells, and PB/PCs in PPs from young UREΔ/Δ and WT mice. **(I)** UMAP visualization of FACS data of CD62L[hi]CD44[low] naïve (T$_N$), CD62L[hi]CD44[hi] central (T$_{CM}$), CD62L[low]CD44[hi] effector (T$_{EM}$) memory, and CD62L[low]CD44[low] pre-effector memory (T$_{pEM}$) T-cell populations in PPs from young UREΔ/Δ and WT mice. **(J)** Numbers of CD4[+] and CD8[+] T cells and T$_{EM}$ and T$_{pEM}$ CD4[+] and CD8[+] subpopulations in PPs from young UREΔ/Δ and WT mice. Each circle represents an individual mouse. *$P < 0.05$; **$P < 0.01$; ***$P < 0.001$; ****$P < 0.0001$ (*t* test).

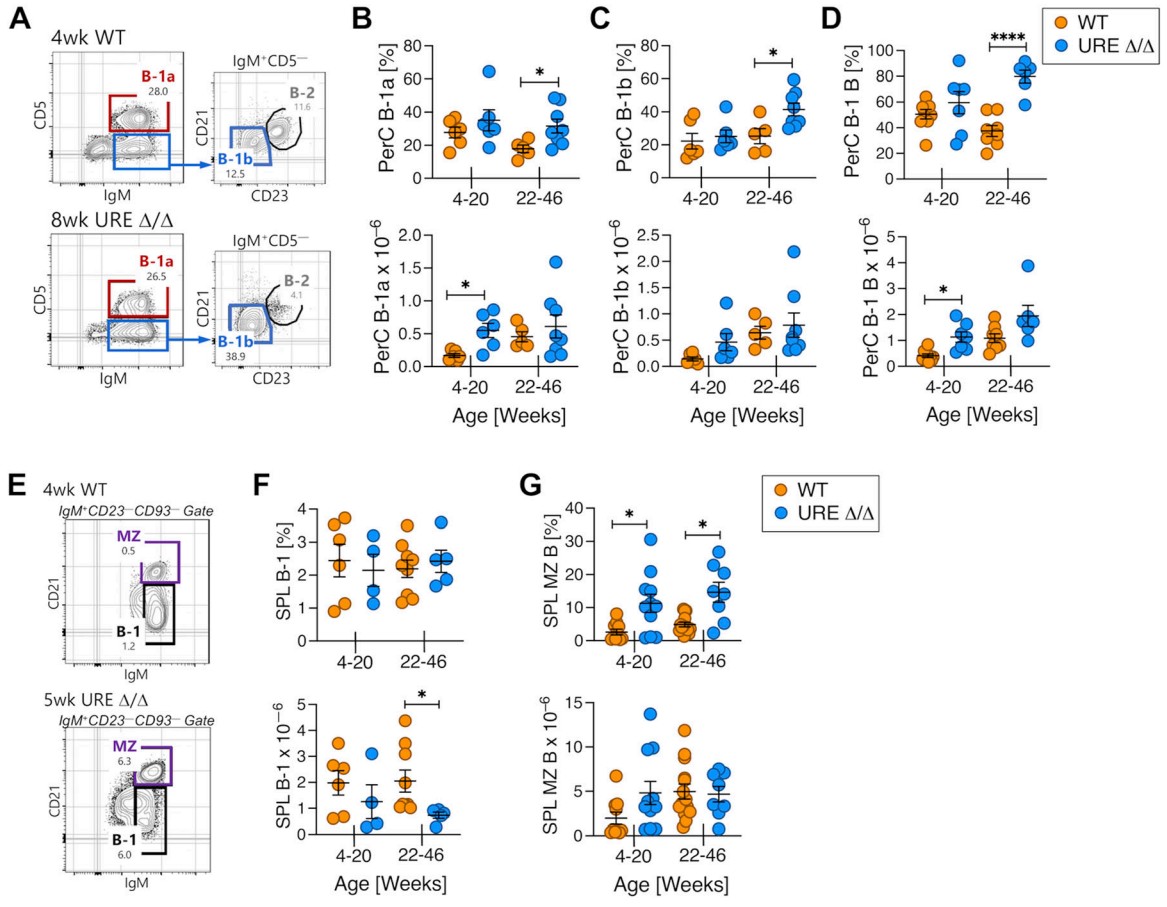

**Figure 4. PU.1^low fetal progenitors efficiently generate innate B cells.**
**(A)** Representative FACS plots showing resolution of B-1a, B-1b, and B-2 B cells in the PerC of UREΔ/Δ and WT mice. **(B)** Frequency and number of B-1a B cells in the PerC of young and middle-aged UREΔ/Δ and WT mice. **(C)** Frequency and number of B-1b B cells in the PerC of young and middle-aged UREΔ/Δ and WT mice. **(D)** Frequency and number of total B-1 B cells in the PerC of young and middle-aged UREΔ/Δ and WT mice. **(E)** Representative FACS plots showing resolution of B-1 and MZ B cells in the SPL of young UREΔ/Δ and WT mice. **(F)** Frequency and number of total B-1 B cells in the SPL of young and middle-aged UREΔ/Δ and WT mice. **(G)** Frequency and number of MZ B cells in the SPL of young and middle-aged UREΔ/Δ and WT mice. Each circle represents an individual mouse. *P < 0.05; ****P < 0.0001 (t test).

blocked, we performed additional analyses to confirm that this was the case.

Our data demonstrated that B-cell development had ceased in UREΔ/Δ mice by middle age because no B-cell progenitors were detected in the bone marrow. A few pro-B or pre-B cells were present in young adult UREΔ/Δ mice, and this can be attributed to the fact transient PU.1^low fetal progenitors may persist through young adulthood before their ultimate loss because of limited self-renewal potential. Similarly, a few ETPs were detected in the thymus of young UREΔ/Δ mice, but de novo thymopoeisis had also ceased by middle age. PU.1 regulates interleukin-7 receptor (IL-7R) expression (DeKoter et al, 2002), and there are reports that thymocytes have extensive self-renewal potential and thymopoiesis can be sustained long-term in the absence of signaling through the IL-7R (Martins et al, 2012; Peaudecerf et al, 2012). However, our data indicate that this does not occur in the UREΔ/Δ thymus because thymopoiesis had ceased in UREΔ/Δ mice by 11 wk after birth. Taken together, our detailed analyses support the conclusion that adult lymphopoiesis is blocked in UREΔ/Δ mice and that the B- and T-cell subsets found in their tissues were generated from transient fetal progenitors.

The analysis of UREΔ/Δ mice demonstrated that in addition to efficiently generating innate B cells, transient fetal progenitors distinguished by their low expression of *Spi1* generated naïve B and T cells in early life. The latter result is consistent with our demonstration of transient fetal waves of B-2 B (Montecino-Rodriguez et al, 2016) and αβ T (Montecino-Rodriguez et al, 2018) cell development in UREΔ/Δ mice. The data further revealed that memory B and T cells derived from one or more of these populations were retained in secondary lymphoid tissues through at least middle age (Fig 7D). These results suggested that the B- and T-cell progeny of transient PU.1^low fetal progenitors can populate secondary lymphoid tissues at numbers sufficient to provide immune protection. That conclusion is consistent with findings that PU.1 expression is not required for normal B-cell function (Polli et al, 2005; Willis et al, 2017).

Nevertheless, secondary lymphoid tissues in UREΔ/Δ mice were not normal in the absence of adult lymphopoiesis, which was necessary to maintain the naïve B- and T-cell compartments. Naïve B-2 B and CD4 and CD8 T-cell progeny of transient PU.1^low fetal progenitors were present in secondary lymphoid tissues from

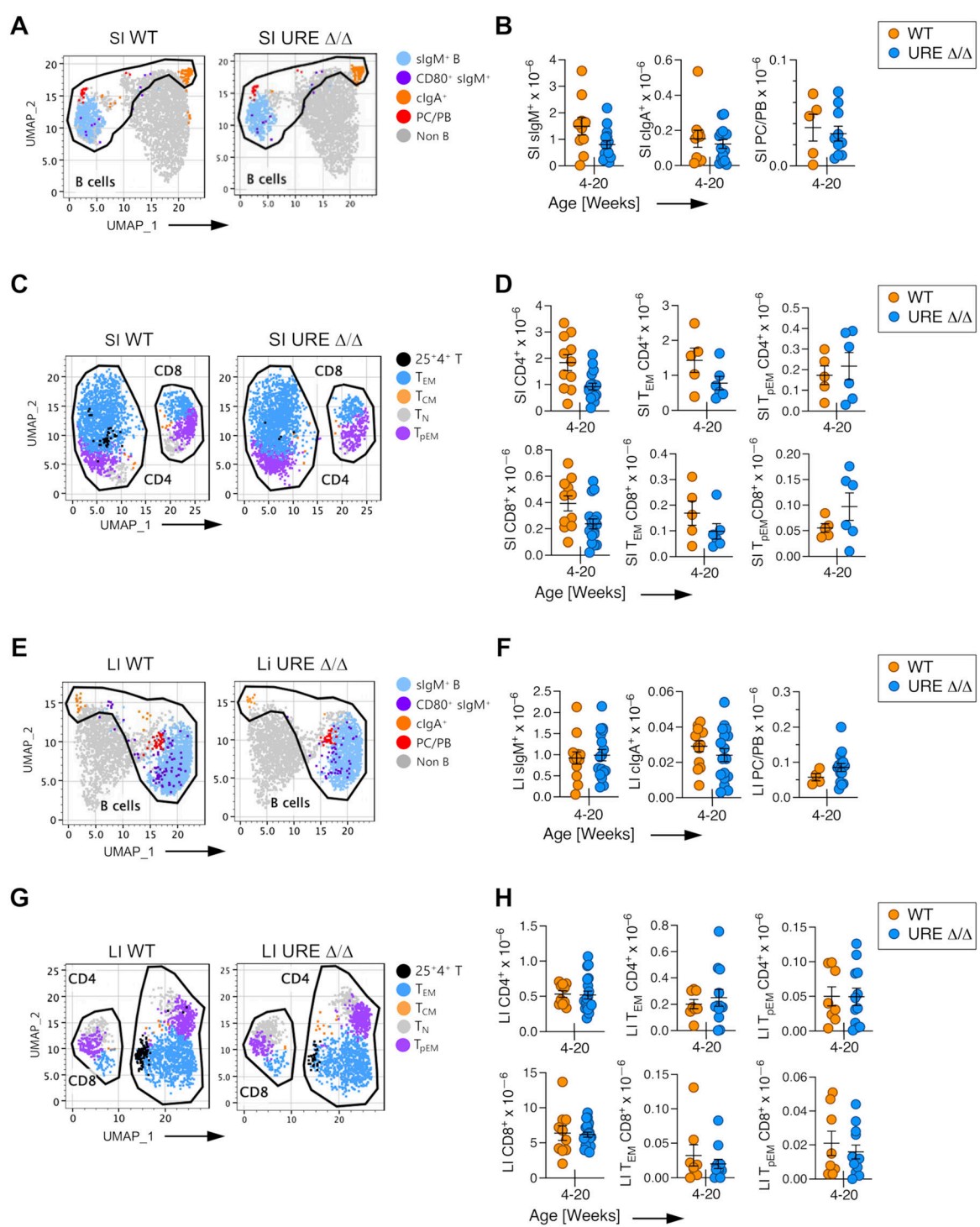

**Figure 5. Activated and memory B and T cells are present in the SI and LI of young UREΔ/Δ mice.**
**(A)** Uniform Manifold Approximation and Projection (UMAP) visualization of FACS data of B-lineage cells in the SI of young UREΔ/Δ and WT mice. **(B)** Numbers of sIgM[+] and cIgA[+] B cells and PB/PCs in SI of young UREΔ/Δ and WT mice. **(C)** UMAP visualization of FACS data of CD62L$^{hi}$CD44$^{low}$ naïve (T$_N$), CD62L$^{hi}$CD44$^{hi}$ central (T$_{CM}$), CD62L$^{low}$CD44$^{hi}$ effector (T$_{EM}$) memory, and CD62$^{low}$CD44$^{low}$ pre-effector memory (T$_{pEM}$) T-cell populations in the SI of young UREΔ/Δ and WT mice. **(D)** Numbers of CD4[+] and CD8[+] T cells and of T$_{EM}$ and T$_{pEM}$ CD4[+] and CD8[+] subpopulations in SI of young UREΔ/Δ and WT mice. **(E)** UMAP visualization of FACS data of B lineage cells in the LI of young UREΔ/Δ and WT mice. **(F)** Numbers of sIgM[+] and cIgA[+] B cells, and PB/PCs in the LI of young UREΔ/Δ and WT mice. **(G)** UMAP visualization of FACS data of CD62L$^{hi}$CD44$^{low}$ naïve (T$_N$), CD62L$^{hi}$CD44$^{hi}$ central (T$_{CM}$), CD62L$^{low}$CD44$^{hi}$ effector (T$_{EM}$) memory, and CD62$^{low}$CD44$^{low}$ pre-effector memory (T$_{pEM}$) T-cell populations in the LI of young UREΔ/Δ and WT mice. **(H)** Numbers of CD4[+] and CD8[+] T cells and of T$_{EM}$ and T$_{pEM}$ CD4[+] and CD8[+] subpopulations in the LI of young UREΔ/Δ and WT mice. Each circle represents an individual mouse. *$P < 0.05$; **$P < 0.01$; ***$P < 0.001$; ****$P < 0.0001$ ($t$ test).

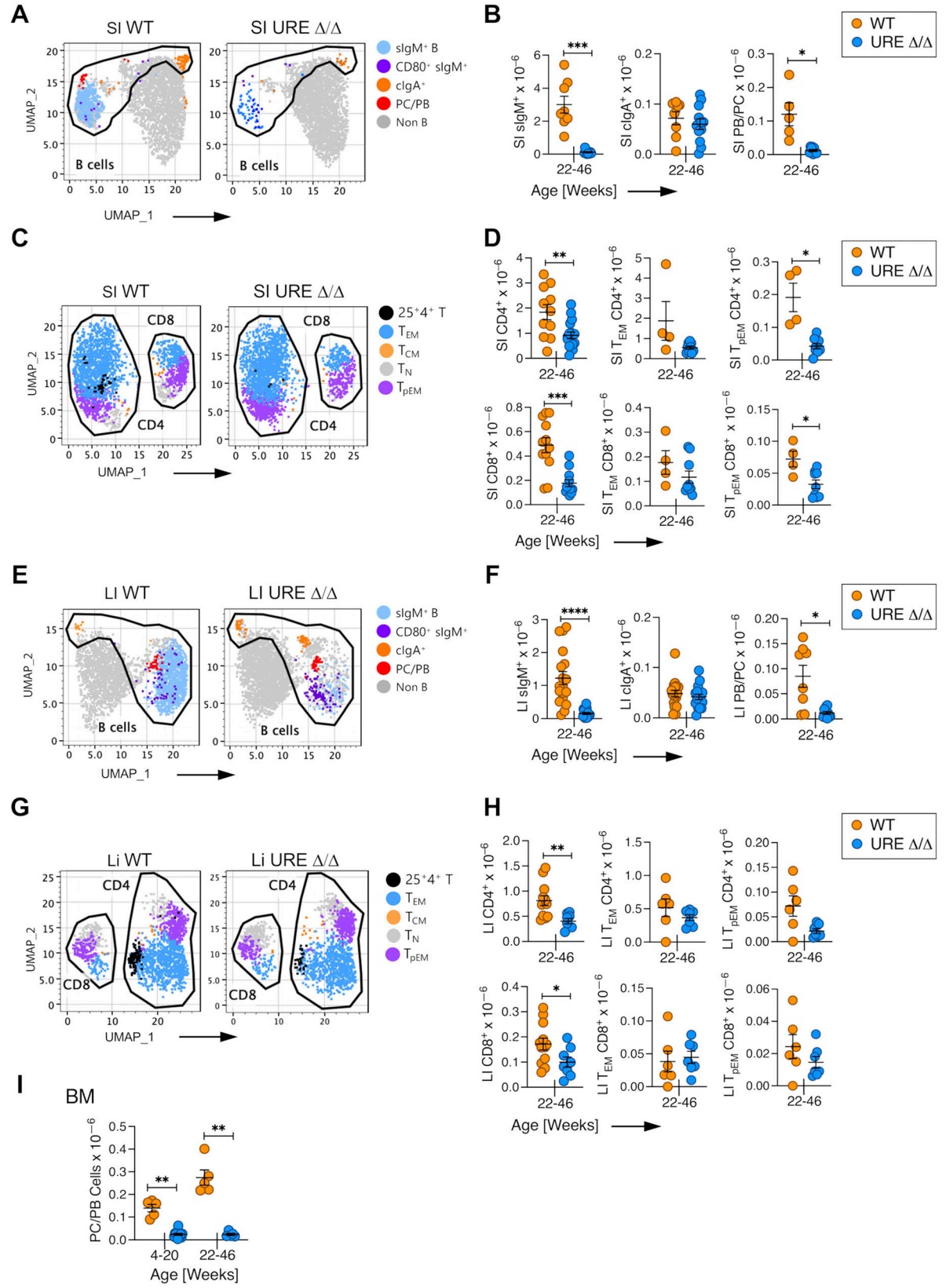

young UREΔ/Δ mice, but their number was significantly lower than in WT animals. As deletion of *Spi1* in mature B (Willis et al, 2017) and T cells (Chang et al, 2009), which do not express detectable PU.1 protein or *Spi1* transcripts (Rothenberg et al, 2019), does not alter their frequency or total number, the reduced number of naïve B and T cells in young UREΔ/Δ mice did not occur because they lacked WT levels of PU.1 expression. Instead, the likely explanation for why naïve B and T cells are maintained at higher numbers in young WT compared with UREΔ/Δ mice is that, in the former animals, waves of lymphopoiesis from both transient PU.1$^{low}$ fetal progenitors as well as stem and progenitor cells that sustain adult lymphocyte production are intact. In contrast, the adult wave of lymphopoeisis is blocked in UREΔ/Δ mice and the transient PU.1$^{low}$ fetal progenitors alone do not generate enough cells to fill the naïve B and T compartments in young animals (Fig 7D). This situation is exacerbated by middle age where all de novo lymphopoiesis has ceased in UREΔ/Δ mice whereas naïve B and T cells are still generated in WT animals. A recent lineage tracing study showing that some waves of fetal lymphopoiesis produced B and T cells present only transiently after birth is in accord with our data (Kobayashi et al, 2023) even though the limited phenotyping in that report did not establish if these were naïve populations, quantification of total cell numbers was not performed, and analysis of a large cohort of middle-aged mice was lacking. Our results demonstrated that, because of their transient nature, PU1$^{low}$ fetal progenitors did not maintain the naïve B-2 and T-cell compartments in adults and that production of these populations is ultimately dependent on lymphocyte development from adult stem and progenitor cells.

Although naïve B and T cells both declined in number between young and middle age in UREΔ/Δ mice, the loss of B-2 B cells was particularly severe. The near complete loss of naïve B-2 B cells is consistent with a previous report indicating that they have a half-life of around 4.5 mo (Hao & Rajewsky, 2001). Thus, in the absence of adult B-2 development as just discussed, the naïve B-2 B cells generated from transient PU.1$^{low}$ fetal progenitors were almost totally lost by middle age. In contrast, a few naïve CD4 and CD8 T cells were present in the spleen of middle-aged UREΔ/Δ mice. It has been proposed that naïve T cells generated in early ontogeny, such as those derived from transient PU.1$^{low}$ fetal progenitors, undergo proliferation because of their residence in a lymphopenic environment (Min et al, 2003) and that this in turn increases their fitness level, thus allowing long-term maintenance (Hogan et al, 2015; Rane et al, 2022). In this case, some naïve T-cell progeny of transient fetal progenitors might survive long-term because of cell intrinsic properties.

In contrast to the transient and limited production of naïve B and T cells, MZ and B-1 B cells were present in UREΔ/Δ mice at WT levels regardless of age. It is known that MZ B cells can be produced form fetal precursors (Carvalho et al, 2001; Beaudin et al, 2016;

Montecino-Rodriguez et al, 2016; Kobayashi & Yoshimoto, 2023), and our data demonstrate that these included PU.1$^{low}$ fetal progenitors. However, the degree to which MZ B cells derived from fetal progenitors are sustained after birth was unknown, and our data indicate that they can be long-lived. Our results further suggest that maintenance of the MZ B-cell compartment is not dependent on ongoing B lymphopoiesis. This conclusion is consistent with a previous study showing that the MZ B-cell compartment was maintained in mice in which adult B-cell development was blocked (Hao & Rajewsky, 2001). Nevertheless, some input into the adult MZ B-cell compartment must normally occur after birth because the adult SPL contains transitional B cells that can generate MZ B cells (Allman & Pillai, 2008; Pillai & Cariappa, 2009).

It is known that B-1 B cells, and B-1a B cells in particular, are efficiently generated from fetal progenitors (Kantor & Herzenberg, 1993; Montecino-Rodriguez & Dorshkind, 2012; Kobayashi et al, 2023), and our data showed that B-1 B cells were present at WT levels in UREΔ/Δ mice through middle age. This was the case even though the initial wave of HSC independent yolk sac B-1 B-cell development is blocked in UREΔ/Δ mice (Montecino-Rodriguez et al, 2016). These results suggested that transient PU.1$^{low}$ fetal progenitors are the source of most B-1 B cells in the adult. In contrast to maintenance of conventional B-2 B cells that require continual input from the bone marrow, B-1 B cells are a self-renewing population (Kantor et al, 1995). Thus, once B-1 B cells are generated, such as from transient fetal progenitors, there is no need for further de novo B-1 B-cell production. This conclusion is in accord with the aforementioned report from Hao and Rajewsky showing that the B-1 compartment is intact in mice with a block in adult B-cell development (Hao & Rajewsky, 2001).

Our conclusion that B-1 B cells in the adult are derived from transient fetal progenitors is at odds with the identification of cells in adult bone marrow with B-1 potential (Berland & Wortis, 2002; Düber et al, 2009; Esplin et al, 2009), and a recent lineage tracing study which concluded that there was extensive postnatal production of B-1a B cells (Vergani et al, 2022). It is important to note that the latter study examined bone marrow from 19 d old mice. However, because some fetal stem and progenitor cells (Bowie et al, 2007), which include CD19$^+$ CD45R(B220)$^-$ B-1 B-cell progenitors (Montecino-Rodriguez et al, 2006), are present in appreciable numbers in the weeks after birth, those fetal progenitors were likely time stamped in the lineage tracing study. In support of this possibility, we showed that HSCs and common lymphoid progenitors from 2.5-wk–old mice efficiently generated B-1 B cells but B-1 potential from stem and progenitor cells was significantly attenuated in 15-wk–old young adult mice (Barber et al, 2011) indicating that marrow cells from older mice have limited B-1 potential.

**Figure 6. PU.1$^{low}$ progenitors generate activated and memory cells in the SI and LI of middle-aged mice.**
**(A)** Uniform Manifold Approximation and Projection (UMAP) visualization of FACS data of B lineage cells in the SI of middle-aged UREΔ/Δ and WT mice. **(B)** Numbers of sIgM$^+$ and cIgA$^+$ B cells and PB/PCs in SI of middle-aged UREΔ/Δ and WT mice. **(C)** UMAP visualization of FACS data of CD62L$^{hi}$CD44$^{low}$ naïve (T$_N$), CD62L$^{hi}$CD44$^{hi}$ central (T$_{CM}$), CD62L$^{low}$CD44$^{hi}$ effector (T$_{EM}$) memory, and CD62$^{low}$CD44$^{low}$ pre-effector memory (T$_{pEM}$) T-cell populations in the SI of middle-aged UREΔ/Δ and WT mice. **(D)** Numbers of CD4$^+$ and CD8$^+$ T cells and T$_{EM}$ and T$_{pEM}$ CD4$^+$ and CD8$^+$ subpopulations in SI of middle-aged UREΔ/Δ and WT mice. **(E)** UMAP visualization of FACS data of B lineage cells in the LI of middle-aged UREΔ/Δ and WT mice. **(F)** Numbers of sIgM$^+$ and cIgA$^+$ B cells, and PB/PCs in the LI of middle-aged UREΔ/Δ and WT mice. **(G)** UMAP visualization of FACS data of CD62L$^{hi}$CD44$^{low}$ naïve (T$_N$), CD62L$^{hi}$CD44$^{hi}$ central (T$_{CM}$), CD62L$^{low}$CD44$^{hi}$ effector (T$_{EM}$) memory, and CD62$^{low}$CD44$^{low}$ pre-effector memory (T$_{pEM}$) T-cell populations in the LI of middle-aged UREΔ/Δ and WT mice. **(H)** Numbers of CD4$^+$ and CD8$^+$ T cells and T$_{EM}$ and T$_{pEM}$ CD4$^+$ and CD8$^+$ subpopulations in the LI of middle-aged UREΔ/Δ and WT mice. **(I)** Number and frequency of plasma cells in the bone marrow of WT and UREΔ/Δ mice. Each circle represents an individual mouse. *P < 0.05; **P < 0.01; ***P < 0.001; ****P < 0.0001 (t test).

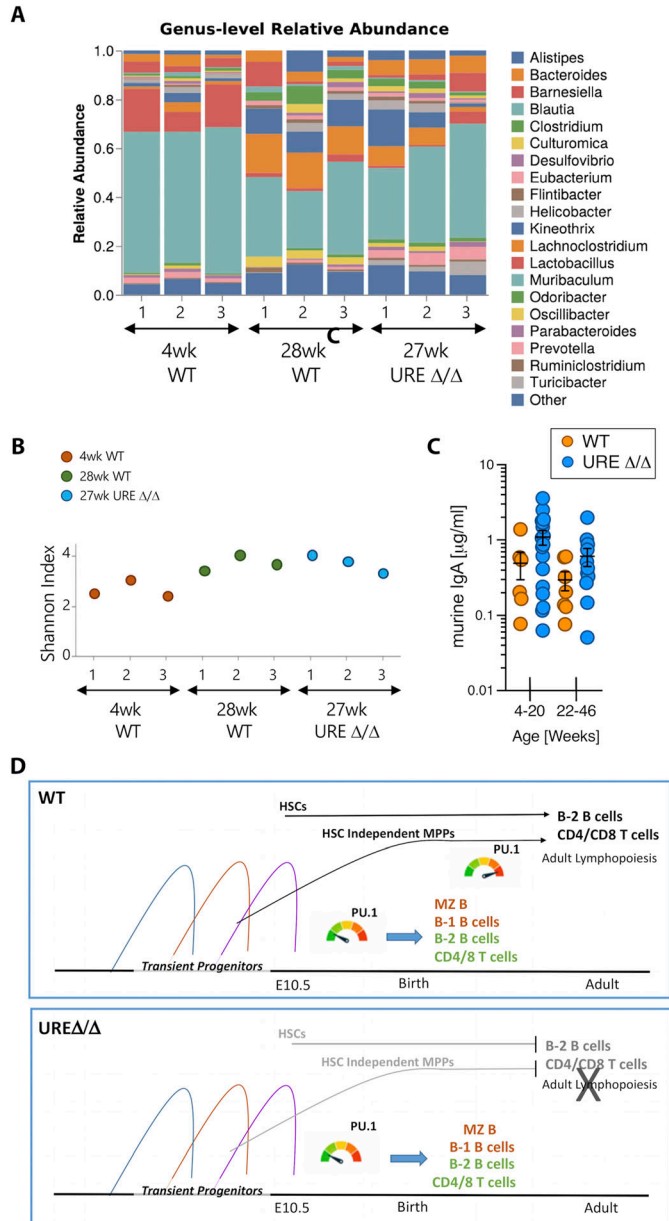

**Figure 7. The lymphoid progeny of transient PU.1^low fetal progenitors maintain gut homeostasis.**
**(A)** Relative abundance chart at the genus taxonomic level of bacteria in fecal samples from UREΔ/Δ and WT mice of the indicated ages. **(B)** Box-plot representation of the Shannon index of diversity relative to species. **(C)** Concentration of IgA in fecal samples from young and middle-aged UREΔ/Δ and WT mice. Each dot indicates an individual mouse. **(D)** Summary figure showing that both the transient fetal and adult waves of lymphopoiesis are intact in WT mice. However, only the transient wave(s) driven by PU.1^low progenitors are intact in UREΔ/Δ mice.

Nevertheless, the possibility that B-1 progenitors in postnatal bone marrow produced some B-1 B cells throughout life cannot be excluded. It is interesting in this regard that, in contrast to the PerC in which the number of B-1 B cells was similar in UREΔ/Δ and WT mice, they were reduced in the SPL of the former animals by middle age. This observation raises the possibility that the B-1 B-cell

compartment in the spleen, but not the peritoneal cavity, is replenished by adult B-1 progenitors after birth.

We observed that a robust compartment of memory/activated B and T cells had established in young and middle-aged UREΔ/Δ mice, particularly in the gut. The intestines are colonized by lymphocytes in utero (Spencer et al, 1986; Golby et al, 2002; Schreurs et al, 2019; Stras et al, 2019), and our data suggested that transient PU.1^low fetal progenitors were a source of these B and T cells. The intestines are a front line immune tissue continually exposed to antigens from the diet and bacteria (Mowat & Agace, 2014), and the infant is exposed to microbial challenge in that tissue immediately after birth. Thus, there is a need to rapidly establish immunity in the gut, and we propose that a major role of the B and T cells generated early in life from transient PU.1^low progenitors is to do so. The fact that there was significant colonization of B and T cells in the SI and LI in young UREΔ/Δ mice is consistent with this view. The data further demonstrated that memory cells derived from PU.1^low progenitors were retained long-term in the intestines. However, the decline in the number of CD8^+ T cells in the LI of middle-aged UREΔ/Δ mice was not as significant as was observed for CD4^+ T cells. Whether this occurred because they are stem-like CD8 T cells that are maintained because of persistent antigen exposure (Lin et al, 2016; Gill et al, 2023; Humblin et al, 2023), which would be encountered in the large bowel, remains to be determined.

Whether the activated, memory, and class-switched cIgA^+ B cells in the intestines of UREΔ/Δ mice were derived from B-1 and/or B-2 B cells requires further study, and this has been a controversial issue. Some reports (Kroese et al, 1989; Murakami et al, 1994; Macpherson et al, 2000; Kunisawa et al, 2007), including two recent lineage tracing studies (Vergani et al, 2022; Tellier et al, 2024), indicated the potential of B-1 B cells to make a contribution to intestinal immunity whereas others indicate that they do not do so (Boursier et al, 2002; Hamada et al, 2002; Thurnheer et al, 2003; Jiang et al, 2004; Roy et al, 2013). We could not optimize a reliable phenotyping protocol to assess whether the cIgA^+ cells we identified in the SI and LI of UREΔ/Δ mice had a B-1 or B-2 B-cell origin. It will ultimately be important to develop approaches to do so to establish the proportion of B cells in the gut that have a B-1 versus B-2 origin.

A limitation of the UREΔ/Δ mouse model is that, in the absence of competition from B and T cells produced in the adult wave of lymphopoiesis, the fetal derived B- and T-activated/memory populations defined herein might establish and survive longer than they would in WT mice in which adult B and T cells are present. However, a direct comparison of our data with the results of Vergani et al (2022) suggests that this is not the case. Those investigators showed that IgA^+ plasma cells derived from fetal/neonatal progenitors were maintained in the SI through at least 18 mo of age; this was the case even though adult lymphopoiesis was intact. However, the frequency of the time stamped IgA^+ plasma cells had declined by ~50% in 12-mo–old mice. We compared the number of IgA^+ cells in the small intestine of 4–20 and 22–46-wk–old UREΔ/Δ mice and found, similar to what Vergani et al reported, there was also a ~50% decline in their number by middle age. Thus, the IgA^+ cells in the SI of UREΔ/Δ mice behaved similarly to what was found in mice in which adult lymphopoiesis was intact. Vergani et al could

not determine if the decline in IgA+ cells they observed was because of the fact that those cells had a limited life span or because there was competition from B cells generated during adult lymphopiesis. As adult B-cell development is blocked in adult UREΔ/Δ mice, our results indicate that there is a gradual loss of IgA+ cells in the small intestine that is mostly because of their life span. Thus, a consistent picture develops when the results from different mouse models are integrated.

Taken together, our results demonstrated that memory and activated B and T cells that are the progeny of transient PU.1[low] fetal progenitors colonized multiple secondary lymphoid tissues, including the intestines, and further suggested that they contributed to the adult immune system through at least middle age. However, UREΔ/Δ mice were generally lymphopenic, indicating that a replete adult immune system is dependent on B and T cells generated from adult stem and progenitor cells. Nevertheless, our data indicated that the adult immune system is constituted by cells derived from progenitors with a differential dependence on the *Spi1* pioneer transcription factor. These results parallel what has been observed for macrophage development where the Myb transcription factor is required for the development of monocytes and macrophages from definitive adult HSCs but is dispensable for the emergence of tissue resident macrophages from transient yolk sac progenitors (Schulz et al, 2012). These observations indicate that the composition of the adult immune system can only be understood from the perspective of layered immune system development and the fact that stem and progenitors arising in transient and sustained waves of development generate B- and T-cell progeny through distinctly regulated genetic pathways.

# Materials and Methods

### Study design

This study was designed to determine the degree to which B and T cells generated in fetal waves of lymphocyte development colonized secondary lymphoid tissues, were retained after birth, and contributed to immunity in the adult. We used the PU.1 deficient Sfpi1tm1.3Dgt/J (UREΔ/Δ) mice because previous reports showed that fetal lymphopoiesis is intact, but adult lymphocyte development is blocked in this strain (Rosenbauer et al, 2006; Montecino-Rodriguez et al, 2016, 2018). Animals were not subjected to experimental interventions except for genotyping by tail clipping during the 1st wk of life. Both male and female mice were included in this study. Our analyses revealed no sex-related effects on the survival of fetal derived B and T cells. Mice included in this study were individually processed and genotyped at the time of sacrifice to confirm their PU1 hypomorphic or WT genotype.

The conclusions in this report are based on analysis of groups of at least eight animals per age tested. This number was determined using the G*Power statistical program (Faul et al, 2007, 2009), which indicated that detection of significant differences of the means within 1.5–2 SD with 95% confidence and an alpha value of 0.05 (probability of type 1 error) for very small hematopoietic populations as determined by flow cytometry would require groups of 8

mice. The results were analyzed for statistical significance using the *t* test included in Prism GraphPad software (Version 10.1.1). Individual data points corresponding to each animal are plotted in our figures. PU.1 hypomorphic and WT control mice ranging in age from 4 to 46 wk of age were used. Subsequent statistical analysis of all the data revealed that PU.1 hypomorphic animals between 4–20 wk of age had a similar immune cell profile that differed from 22- to 46-wk–old animals. Based on this, the data are presented based on whether the mice were young (4–22 wk old) or middle aged (22–46 wk old). The investigators who performed the experiments were not blinded.

### Mice

Breeding pairs of Sfpi1tm1.3Dgt/J (UREΔ/Δ) mice were originally obtained from the Jackson Laboratory and were backcrossed with C57BL/6 (B6) mice for at least seven generations in the UCLA Division of Laboratory Animal Medicine. Although the original publication that described the generation of UREΔ/Δ mice indicated that they developed various leukemias (Rosenbauer et al, 2006), this was only observed in ~5% of mice in our colony; animals with signs of disease were excluded from analysis. Age matched UREΔ/Δ heterozygotes and C57BL/6 mice were used interchangeably as WT controls based on our previous studies which had shown that the immune system of these animals did not differ significantly (Montecino-Rodriguez et al, 2016). All protocols were approved by the UCLA Institutional Animal Care and Use Committee. Genotyping of UREΔ/Δ mice was performed by amplification of cDNA samples extracted from neonatal tail clips using the Qiagene DNeasy Blood and tissue kit and the Standard PCR Assay—Sfpi1<tm1.3Dgt> protocol # 22510 as recommended on the Jackson Laboratory website.

### Harvest and preparation of tissues

Bone marrow, SPL, MLNs, and PerC-cell suspensions were harvested and processed as previously described (Min et al, 2006; Montecino-Rodriguez et al, 2006). Cells were isolated from SI and LI lamina propria using a protocol adapted from Sheridan and Lefrançois (2012). Whole mouse gut was harvested by cutting it at the pylorus and at the anus and cleaned from mesenteric fat. The tissue was then separated into SI and LI, which included the cecum, and placed in cold Ca++Mg++ free PBS supplemented at 1% with heat inactivated (hi) FCS. PPs were excised from SI using a microscope before proceeding with the subsequent steps. SI and LI were opened longitudinally, most digestive debris was gently scrapped off using the edge of a glass slide, and any remaining debris was removed by gently shaking of the tissues in PBS supplemented with cold Ca++Mg++ free PBS 1% hi FCS. These intestinal tissues were then cut in ~1 cm fragments and incubated with a large volume of phenol red and bicarbonate free RPMI 1460 (Gibco) supplemented with 10% hi FCS, 15 mM Hepes (Gibco) 5 mM EDTA (Sigma-Aldrich) and 1 mM DTT (Thermo Fisher Scientific) for 30 min at 37°C in shaker set at 200 rpm to remove epithelial cells and mucus. The SI and LI fragments were then sedimented by gravity and the RPMI/EDTA/DTT supernatant was discarded, rinsed with large volumes of cold Ca++Mg++ free PBS, drained and minced in a solution at 200 U/ml of collagenase type 1A (Sigma-Aldrich) in phenol red and bicarbonate

free RPMI 1460 10% hi FCS and 15 mM Hepes and digested for 30 min at 37°C in shaker set at 200 rpm. The SI and LI slurries were run through a syringe fitted with an 18 G needle, filtered using a 70 $\mu$m nylon mesh. The released cells were then washed with cold Ca$^{++}$Mg$^{++}$ free PBS and centrifuged at 400$g$ for 5 min at RT. Lymphoid cells were then enriched by use of a discontinuous 40%:70% Percoll gradient.

PPs were incubated in RPMI/EDTA/DTT for 15 min at 37°C to remove epithelial cells and mucus. They were then sedimented by gravity, rinsed with cold Ca$^{++}$Mg$^{++}$ free PBS and digested for 15 min at 37°C in a solution at 200 U/ml of collagenase type 1A (Sigma-Aldrich) in phenol red and bicarbonate free RPMI 1460 10% hi FCS and 15 mM Hepes. PPs slurries were run through a syringe fitted with an 18 G needle, filtered using a 70 $\mu$m nylon mesh and washed with cold Ca$^{++}$Mg$^{++}$ free PBS and centrifuged at 400$g$ for 5 min at RT.

Red blood cells in cell suspensions were lysed using 1X ACK lysis buffer on ice for 3 min. The cells were then counted with a hemocytometer and cell viability determined by eosin dye exclusion.

### Flow cytometry

All staining procedures were performed in cold Ca$^{++}$Mg$^{++}$ free PBS. Samples were first incubated with a CD16/32 antibody to block non-specific binding of antibodies to cells via Fc receptors. All antibodies used are listed in Table S1. For surface staining, cells were incubated on ice for 30 min with the appropriate dilution of antibodies. Unbound antibodies were washed from cells with cold Ca$^{++}$Mg$^{++}$ free PBS. To be able to exclude dead cells during analyses of samples for intracellular staining, cells were first pre-incubated with the Fixable Viability Dye eFluor 780 (Thermo Fisher Scientific), as recommended by the manufacturer protocol. Subsequently, the cells were stained for surface antigens and fixed at 4°C for 20 min with 0.5% paraformaldehyde in Ca$^{++}$Mg$^{++}$ free PBS. The cells were then permeabilized for 15 min with cold 1x BD Perm/Wash Buffer in the presence of a CD16/32 blocking antibody and then stained for anti-murine IgA at 4°C overnight. Unbound antibodies were washed from cells with 1x BD Perm/Wash and cells were resuspended in an appropriate volume of cold Ca$^{++}$Mg$^{++}$ free PBS for flow. Stained samples were run on a LSRII (BD Biosciences) located in the Broad Stem Cell Research Center (University of California at Los Angeles). Frequencies of cell populations were determined using the FlowJo Software v10 samples. To compare the cellular composition of similar tissues samples across multiple ages, we analyzed them using the unsupervised clustering and nonlinear dimensionality reduction Uniform *Manifold* Approximation and Projection (UMAP) plugin developed by McInnes et al (2020 *Preprint*) available in the FlowJo Software v10 and the samples were processed as recommended (Ujas et al, 2023) before running the analyses.

We used standard flow best practices to minimize variations between samples during acquisition: all samples were run on the same LSRII instrument with identical parameter settings to control for fluorescence intensity variation between runs. We also followed the recommended best practice for UMAP analyses: identical number of acquired cells from WT and UREΔ/Δ samples were concatenated by age and tissue before running the UMAP algorithm (FlowJo10). In this case, flow samples that included too low events

or out of average range of fluorescence intensities for the parameters analyzed were excluded.

### 16S microbiome sequencing

Three fecal pellets per mouse were harvested from UREΔ/Δ or WT mice of different age and frozen at—20°C until sequencing. DNA extraction and sequencing from fecal samples were performed by Charles River. DNA was isolated from submitted samples using optimal sample-dependent column or magnetic purification kit. Recovery yield and DNA quality was determined by fluorometric analysis. DNA concentration was adjusted to specifications and amplified using broadly reactive 16s rRNA primers spanning the V3 and V4 regions. Resulting amplified PCR products were analyzed for quantity and correct product size then purified and amplified with primers containing unique sample nucleotide barcodes. PCR products quality and quantity were further analyzed by SYBR green qPCR. All samples were pooled and adjusted to a normalized concentration. The DNA library pool was denatured with sodium hydroxide, normalized to optimal loading concentration, and combined with PhiX control. Extended read lengths up to 2 × 300 bp was used for cluster generation and sequencing. Following the sequencing run, the sequence data were de-multiplexed based on the nucleotide barcode and compared with the One Codex Targeted Loci Database for taxonomy identification and subsequent alpha and beta diversity analysis. All raw sequences were analyzed by Charles River Laboratories bioinformaticians using the One Codex Targeted Loci Database (TLDB)1 that contains 250,000 curated gene records. To analyze samples against the Targeted Loci Database, every read was aligned with high sensitivity (using SNAP) to identify the best alignments to the database. Each read was assigned to the most specific taxonomic grouping that the data support. Abundance-based filtering to minimize the number of false positive assignments that may be introduced by sequencing error was then performed. As amplification using primers to the V3 and V4 regions may not allow reliable quantification of differences at the species level (Johnson et al, 2019), bacterial identification was limited to taxa at the genus level. The raw results of these analyses are provided in the Supplemental Material.

### Fecal IgA quantification by sandwich ELISA

Three fecal pellets per mouse were harvested from the anal canal of UREΔ/Δ and WT mice and resuspended in 500 $\mu$l of sterile Ca$^{++}$Mg$^{++}$ free PBS in 1.5 ml Eppendorf tubes. The samples were well mixed and incubated for 4–5 h at RT. The samples were then vortexed and fecal debris was sedimented by centrifugation at 20,000$g$ for 10 min at 4°C. The supernatants were then transferred to new Eppendorf tubes and stored at –20°C until they were tested for IgA concentrations using a standard double sandwich ELISA. Briefly, wells of microtiter plates (NUNC Immunoplate Maxi Sorp, Denmark) were coated with 50 $\mu$l/well of goat anti anti-mouse IgA overnight at 4°C. The plates were washed with Mg$^{++}$ PBS 0.5% Tween 20 and Mg$^{++}$ PBS and subsequently blocked with Mg$^{++}$ PBS supplemented with 2% BSA (Thermo Fisher Scientific) for 2 h at 37°C. Duplicate samples of 50 $\mu$l/well of supernatant were distributed in wells and incubated for 1 h and 30 min at 37°C. The plates were

washed with Mg$^{++}$ PBS 0.5% Tween 20 and Mg$^{++}$ PBS and bound IgA was detected using 50 $\mu$l/well appropriate dilutions of goat anti-mouse IgA alkaline phosphatase-conjugated antibody in Mg$^{++}$ PBS 0.5% Tween 20 0.5% BSA. After final washes, 100 $\mu$l of $p$-Nitrophenyl phosphate (PNPP, Sigma-Aldrich) at 0.1 mg/ml in 1x Diethaloamine buffer (Thermo Fisher Scientific) were added per well as substrate. Optical density was read after a 30 min incubation at 37°C using a plate reader fitted with 405 nm filter. IgA concentrations in the supernatants were determined using a standard curve obtained using serial dilutions of known concentrations of murine IgA. The monoclonal antibodies used to detect murine IgA in this assay are listed in Table S1.

### Statistical analysis

Data are expressed as a mean ± SEM, as indicated in the figure legends. Differences between groups were tested using a two-tailed, unpaired $t$ test. As noted in the study design, the number of animals processed was determined based on the G*Power statistical program (Faul et al, 2007, 2009). The data are presented so that information about each individual mouse processed can be viewed.

## Data Availability

All data needed to evaluate the conclusions are presented in the article or the Supplementary Materials. All mouse strains are available from the Jackson Laboratory.

## Supplementary Information

## Acknowledgements

This work was supported by grants from the NIH (AI021256) and the Chron's and Colitis Foundation (#1654). The Jonsson Comprehensive Cancer Center flow core is supported by NIH grant CA16042.

### Author Contributions

E Montecino-Rodriguez: conceptualization, formal analysis, investigation, methodology, and writing—original draft, review, and editing.
O Estrada: methodology.
K Dorshkind: conceptualization, formal analysis, funding acquisition, investigation, methodology, and writing—original draft, review, and editing.

### Conflict of Interest Statement

The authors declare that they have no conflict of interest.

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
