## [Reviewer comments · Life Science Alliance]

Life Science Alliance

Transient PU.1 Low Fetal Progenitors Generate Lymphoid Progeny that Contribute to Adult Immunity

Encarnacion Montecino-Rodriguez, Oscar Estrada, and Kenneth Dorshkind

DOI: <https://doi.org/10.26508/lsa.202402629>

Corresponding author(s): Kenneth Dorshkind, David Geffen School of Medicine at UCLA

Review Timeline:

Submission Date:	2024-01-30
Editorial Decision:	2024-03-28
Revision Received:	2024-05-15
Editorial Decision:	2024-05-22
Revision Received:	2024-05-22
Accepted:	2024-05-23

Transaction Report:

March 28, 2024

Re: Life Science Alliance manuscript #LSA-2024-02629-T

Dr. Ken Dorshkind
David Geffen School of Medicine at UCLA
Department of Pathology and Laboratory Medicine
Pathology and Laboratory Medicine
19833 Le Conte Avenue
Los Angeles, California 90095

Dear Dr. Dorshkind,

Thank you for submitting your manuscript entitled "Transient Fetal Progenitors With Low Dependence on PU.1 Generate Lymphoid Progeny that Contribute to the Adult Immune System" to Life Science Alliance. The manuscript was assessed by expert reviewers, whose comments are appended to this letter. We invite you to submit a revised manuscript addressing the Reviewer comments.

Thank you for this interesting contribution to Life Science Alliance. We are looking forward to receiving your revised manuscript.

Sincerely,

Eric Sawey, PhD
Executive Editor
Life Science Alliance
<http://www.lsa-journal.org>

B. MANUSCRIPT ORGANIZATION AND FORMATTING:

Reviewer #1 (Comments to the Authors (Required)):

This paper uses multiparameter flow cytometry analysis of diverse populations of B and T lymphoid cells at ages ranging over nearly half of a mouse's maximum lifetime to compare wildtype mice with mice that apparently only make lymphoid cells in fetal life from special PU.1-low precursors, namely homozygous germline deltaURE/deltaURE mice. The results show that despite the severe inhibition of new lymphocyte production in the mutant animals starting around birth, they continue to maintain innate-like, effector-like, and memory-like B and T cell populations and they continue to maintain normal gut bacterial population homeostasis. Thus, the animals are much healthier than one might imagine in light of their shortage of new lymphoid production. The results are notable, although some aspects of the presentation make them a little hard to evaluate. The Discussion is very scholarly and explicit in relating the new results presented to ongoing debates in the field. A weakness of the paper in its current form is that it does not include any attempt to test precursor-product relationships directly. The authors interpret the phenotypes of the intact, unmanipulated mutant mice as exclusively coming from cells of fetal origin and argue that the ability to avoid cell transfer experiments is a strength of the system. However, it is not clear whether the previous studies of these mutants are quantitatively strong enough to be certain about this without any more direct evidence, especially when extrapolating to the cells accumulating in much older mice than before. The evidence shown is clearly interesting and valuable, but the paper could be stronger with some possible corrections and a few additional experiments or citations of previous experiments.

1. In the authors' previous papers about the delta URE homozygous mice, they concentrated on comparing fetal with young adult mice. In this paper, they look at mice of much more advanced age. It is interesting that even these older mice are so healthy, considering that the original report about the delta URE homozygous mutants (Rosenbauer et al., 2006) made a major point about the T leukemias and acute myelogenous leukemias that showed up in the line at high prevalence. The ages of onset for the leukemias described in the original Rosenbauer report overlapped extensively with the ages of the mice that are studied in this manuscript, but nothing is said here about malignancy one way or another. Could the authors please comment explicitly? If the malignancy no longer appears (for example in the current congenic strain background used here), it would be valuable to state that.

2. The T cell abnormalities in the aged mutants seem to be more dramatic than the B cell effects, with Fig. 1D showing a nearly total loss of T cells except for CD8+ memory T cells in the spleens of mutant middle-aged mice. However, there are some questions about the T cell scoring. In Fig. S2C, the wildtype CD4+ T cells appear to have an unusually minimal population of naïve T cells and a huge preponderance of T effector memory cells. Is this normal, or could the axes be reversed? Showing actual flow cytometry for the younger samples would be helpful to resolve this. Also, where a more complete age range is compared in Fig. 2E, the y axis numbers do not add up. Presumably at least one of the subsets is present at 10x higher numbers than the y axes indicate, if the different populations are to add up to the total numbers of CD4 or CD8 cells given in the left panels of Fig. 2E. If the mis-calibrated one is the naïve T cells, then the panels in Fig. S2C are probably mislabeled.

3. The main concern is that steady-state populations in the mutants up to a year after birth are interpreted as though it is certain that lymphocyte production has stopped completely at birth. However, what if the rate of lymphocyte production drops a lot, but is not totally eliminated in the mutants? There is certainly some compensatory proliferation occurring due to lymphopenia, as there must be massive expansion of the output of a tiny number of fetal precursors to generate any of the populations seen. But does this have to apply only to purely fetal origin cells? Over a 40 week period, could a low leakage rate of per-cell precursor activity remaining in the mutant adult bone marrow contribute to the accumulated cells as well? One could also worry that if any lymphoid precursors did still exist in postnatal mutants, their low PU.1 could alter their cell surface phenotype (e.g. causing loss of IL-7R and Flt3) so that they might not be easily identified. It could easily be beyond the scope of this paper to answer all these questions fully, but it still seems worth acknowledging them in interpreting the results shown here.

4. Although the unmanipulated animals do offer an advantage, it still seems that the paper could be helped by any data from experiments in which long-term chimeras of mutant fetal vs. early postnatal precursors were tested many months after reconstitution. If there is already a published experiment that already compares the outputs of mutant fetal, wildtype fetal, and wildtype and mutant early postnatal cells quantitatively in adoptive transfer experiments, please cite it explicitly. This could help

to put an upper bound on the amount of postnatal production.

5. Adoptive transfer evidence would also help to resolve whether the subset distributions seen in the older mutant animals are determined cell autonomously all the way out to 9-11 months of age. This seems like a natural question, especially in these aging animals where myeloid cells should be abnormal and dendritic cells severely reduced due to the low PU.1 expression in the animals as a whole. The huge loss of CD4+ T cells at these older ages also seems possible to contribute to effects on the B cell population as the animals age. Again, if results for these tests already exist, please discuss them more explicitly.

Minor points:

1. There have been several reports from the Rudd group about fetal and early postnatal CD8 T cells labeled by a time-stamping method. These have identified early-generated cells which also persist long into adulthood and have distinctive effector function profiles. In principle, those authors should be looking at some of the same cells as Montecino-Rodriguez et al. Are the phenotypes of the delta URE homozygous CD8+ T cells consistent?

2. The altered subset distributions of the mutant PU.1-low cells are interpreted as being affected mostly by the types of progenitors they came from. But is it certain that the reduced levels of PU.1 in the delta URE homozygous B cells are not continuing to make them functionally abnormal as mature cells?

3. Reference Vergani et al., 2022, distinguishes between fetal and early postnatal precursors, but still agrees with Montecino-Rodriguez et al. that all these precursors are very different from later lymphoid precursor populations in fully mature adult mouse bone marrow. Here, the authors strongly argue that the delta URE model shows that the relevant precursors have to be fetal. But is there strong evidence that the real cutoff of developmental capability is exactly at birth for delta URE precursors, or whether it could occur some time slightly later? That is a slightly different question than the one the authors previously addressed when they showed the ontogeny of naturally PU.1-low cells in wildtype mice.

4. Could the authors capitalize the acronyms "SI" and "LI" for small intestine and large intestine, in the text and figures?

5. Use of UMAP dimensional reductions for multiparameter flow cytometry is clever and may inspire other authors to use a similar display. However, there are recognized clues to normal or abnormal status of cells based on particular pattern distortions of the flow biplots in certain pathological situations, and the UMAP plots obscure these signs. It is a shame that the authors did not include more typical displays of flow cytometry data comparing the wildtype and mutant cells: most of it is in Fig. S2, which only shows wildtype data. Additional supplementary figures with more examples comparing wildtype and mutant phenotypes at the same time point would be helpful.

Reviewer #2 (Comments to the Authors (Required)):

The manuscript by Montecino-Rodriguez et al reports the analysis of a mouse that carries a mutation in Pu-1 that drastically reduces expression of this transcription factor, required for lymphocyte development. The authors had previously shown that B and T cell production was maintained in embryo and perinatal progenitors (Montecino-Rodriguez et al 2016) but was strongly reduced from adult progenitors, although the first B1 B cells was strongly reduced in mutant mice. Interestingly the authors also reported that post-natal thymopoiesis was maintained such that 7-week old mutants still exhibited thymic activity although reduced (Montecino-Rodriguez et al 2018). The present report is designed to show that the fetal lymphoid progenitors contribute to adult immunity and intestinal microbioma homeostasis. To illustrate that adult lymphopoiesis is strongly reduced, numbers of T and B cells in the spleen and thymus are low with few naïve lymphocytes being found whereas activated memory cells are at levels close to normal. This finding suggests indeed adult lymphopoiesis is reduced although residual production may occur because FigS1 shows the presence at very low levels of B220+IgM- cells in adult BM and the thymus as mentioned above is active until 7 weeks of age at least. So there is no formal evidence that the activated peripheral cells originate from embryonic and perinatal lymphocytes rather than from low numbers of lymphocytes produced throughout life. Evaluating the presence of N sequences in the V-D-J rearrangements of peripheral B and T cells might contribute to clarify this issue since embryonic lymphocytes exhibit poor repertoire diversity. Another potential limitation is the absence of adult competitors in the experimental setting. It is not evident that in the presence of diverse population of B and T cells the lymphocytes originated perinatally would persist and contribute to adult immunity. A competitive reconstitution experiment might provide additional evidence that these embryonic lymphocytes persist in normal animals. The work is otherwise well performed. However, the claim that the activated cells of embryonic origin contribute to tissue homeostasis is overinterpreted and could simply result from heterogeneity in the samples. The differences at week 27-28 is less impressive than between WT weeks 4 to 27. Can the authors put some quantification in this analysis? Otherwise, I would recommend turning down this claim. It has been previously shown that in the absence of thymic input thymopoiesis can persist for long periods of time (Peaudecerf et al J Exp Med 2012; Martins et al J Exp Med 2014). The authors might like to discuss their data in this context. Another confusing element is that in the schematic representation of hematopoiesis the embryonic/perinatal lymphocyte production is HSC independent although in Montecino-Rodriguez et al 2016 it is stated that the HSC independent wave of yolk sac origin that contributes to the first B1 cells is missing in these mutants. This requires clarification.

May 15, 2024

Eric Sawey, Ph.D.
Executive Editor
Life Science Alliance

Dear Dr. Sawey:

Thank you for sending the reviews of our manuscript, entitled "Transient Fetal Progenitors With Low Dependence on PU.1 Generate Lymphoid Progeny that Contribute to the Adult Immune System (LSA-2024-02629-T)" and the opportunity to submit a revised manuscript. The comments from both reviewers were constructive, and the revised manuscript includes additional data and discussion points to address them. Please note that the revisions in the manuscript are indicated in blue font. In addition, due to character limitations we had to change the title to "Transient PU.1 Low Fetal Progenitors Generate Lymphoid Progeny that Contribute to Adult Immunity".

Reviewer 1

The issue of whether adult lymphocyte development was blocked in adult PU.1 deficient URE Δ/Δ mice was raised in several comments. For example:

General Comment: *A weakness of the paper in its current form is that it does not include any attempt to test precursor-product relationships directly. The authors interpret the phenotypes of the intact, unmanipulated mutant mice as exclusively coming from cells of fetal origin and argue that the ability to avoid cell transfer experiments is a strength of the system. However, it is not clear whether the previous studies of these mutants are quantitatively strong enough to be certain about this without any more direct evidence, especially when extrapolating to the cells accumulating in much older mice than before.*

Point 3. *The main concern is that steady-state populations in the mutants up to a year after birth are interpreted as though it is certain that lymphocyte production has stopped completely at birth. However, what if the rate of lymphocyte production drops a lot, but is not totally eliminated in the mutants? There is certainly some compensatory proliferation occurring due to lymphopenia, as there must be massive expansion of the output of a tiny number of fetal precursors to generate any of the populations seen. But does this have to apply only to purely fetal origin cells? Over a 40 week period, could a low leakage rate of per-cell precursor activity remaining in the mutant adult bone marrow contribute to the accumulated cells as well? One could also worry that if any lymphoid precursors did still exist in postnatal mutants, their low PU.1 could alter their cell surface phenotype (e.g. causing loss of IL-7R and Flt3) so that they might not be easily identified. It could easily be beyond the scope of this paper to answer all these questions fully, but it still seems worth acknowledging them in interpreting the results shown here.*

Point 4. *Although the unmanipulated animals do offer an advantage, it still seems that the paper could be helped by any data from experiments in which long-term chimeras of mutant fetal vs. early postnatal precursors were tested many months after reconstitution. If there is already a published experiment that already compares the outputs of mutant fetal, wildtype fetal, and wildtype and mutant early postnatal cells quantitatively in adoptive transfer experiments, please cite it explicitly. This could help to put an upper bound on the amount of postnatal production.*

Point 5. *Adoptive transfer evidence would also help to resolve whether the subset distributions seen in the older mutant animals are determined cell autonomously all the way out to 9-11 months of age. This seems like a natural question, especially in these aging animals where myeloid cells should be abnormal and dendritic cells severely reduced due to the low PU.1 expression in the animals as a whole. The huge loss of CD4+ T cells at these older ages also seems possible to contribute to effects on the B cell population as the animals age. Again, if results for these tests already exist, please discuss them more explicitly.*

These comments, along with comment #1 from Reviewer 2, raised the issue of whether adult lymphopoiesis is blocked in URE Δ/Δ mice. Reviewer 1 suggested that we transplant stem and progenitor cells from fetal and adult URE Δ/Δ mice into recipients to confirm the absence of lymphoid potential. However, such experiments would not provide clear answers, because PU.1 deficient stem cells do not efficiently engraft in the bone marrow following transplantation (*Blood* 106:1590, 2005). Thus, if we transplanted total bone marrow or purified stem/progenitor cells harvested from URE Δ/Δ mice into appropriately conditioned syngeneic recipients and observed no lymphoid reconstitution, we could not be sure if this was due to their lack of lymphoid potential or to poor reconstitution.

Demonstrating that B and T cell progenitors are absent from unmanipulated, adult URE Δ/Δ mice is the ultimate proof that lymphopoiesis is blocked due to their PU.1 deficiency. Therefore, we performed extensive new phenotyping experiments to quantify B cell progenitors in the marrow and T cell progenitors in the thymus of different aged URE Δ/Δ mice to extend our previous analyses demonstrating that HSCs and MPPs isolated from URE Δ/Δ mice could not generate B and T lymphoid progeny (*Immunity* 45:527, 2016; *J Immunol* 200:2046, 2018). Please note that identification of B and T cell progenitors does not rely on expression of Flt3 or the IL-7 receptor. The data are presented in two new supplementary figures.

Figure S1: B cell production. We performed a detailed phenotypic analysis of B cell development in young and middle aged URE Δ/Δ mice using the scheme described by Hardy (Supplemental Figure 1A). The results confirmed that adult B cell development was blocked in URE Δ/Δ mice. A few B cell progenitors were present in young URE Δ/Δ mice, but at levels below that in WT animals. This is not surprising because an abrupt loss of fetal derived stem/progenitor cells does not occur at birth (please note that we never stated that this occurred). Instead, transient fetal progenitors can persist into young adulthood (see *PNAS* 104:5878, 2007; *Nature Immunol.* 7:293, 2006). The important point to stress is that **no** B cell progenitors were observed in middle aged URE Δ/Δ mice while they were easily detected in WT mice. These data confirm that B lymphopoiesis from adult progenitors is blocked in URE Δ/Δ mice.

Figure S2: T cell production. We quantified early T lineage progenitors (ETPs) and their progeny in the thymus of URE Δ/Δ mice using the gating strategy shown in the (Supplemental Figure 2B). A major decline in thymocyte number was already evident in 4 week old URE Δ/Δ mice, and by 11 weeks after birth few thymocytes were present. Recovery in older animals was close to zero, and ETP and double negative thymocytes were not present. As noted in the revised text, this sharp loss of cells in the thymus of URE Δ/Δ mice **is not** what is observed during thymic involution in WT mice, where there is a gradual loss of thymocytes over time and progenitor populations are still present. Thus, adult thymopoiesis is also blocked in URE Δ/Δ mice.

Please note that we analyzed at least 6 WT and 6 URE Δ/Δ mice per time point with identical results to what is shown in the supplemental figures. Thus, the FACS plots truly represent the block in adult lymphopoiesis in URE Δ/Δ mice. These points are noted in the supplemental figure legends. We would be happy to include figures S1 and S2 in the main text instead of as supplemental figures if the reviewers so request.

Additional Main Comments:

Comment 1. In the authors' previous papers about the delta URE homozygous mice, they concentrated on comparing fetal with young adult mice. In this paper, they look at mice of much more advanced age. It is interesting that even these older mice are so healthy, considering that the original report about the delta URE homozygous mutants (Rosenbauer et al., 2006) made a major point about the T leukemias and acute myelogenous leukemias that showed up in the line at high prevalence. The ages of onset for the leukemias described in the original Rosenbauer report overlapped extensively with the ages of the mice that are studied in this manuscript, but nothing is said here about malignancy one way or another. Could the authors please comment explicitly? If the malignancy no longer appears (for example in the current congenic strain background used here), it would be valuable to state that.

The reviewer correctly notes that the Rosenbauer paper that described the generation of URE Δ/Δ mice reported that they developed leukemia. We observed this in 5% of mice at most. We have maintained URE Δ/Δ mice in our colony since 2015, so one possibility is that the trait(s) responsible for malignant hematopoiesis were lost over time. In any case, the revised Methods section where the mice are described notes that the high incidence of leukemia that was described in the Rosenbauer et al. report was not observed in our colony.

Comment 2. The T cell abnormalities in the aged mutants seem to be more dramatic than the B cell effects, with Fig. 1D showing a nearly total loss of T cells except for CD8⁺ memory T cells in the spleens of mutant middle-aged mice. However, there are some questions about the T cell scoring. In Fig. S2C, the wildtype CD4⁺ T cells appear to have an unusually minimal population of naïve T cells and a huge preponderance of T effector memory cells. Is this normal, or could the axes be reversed? Showing actual flow cytometry for the younger samples would be helpful to resolve this. Also, where a more complete age range is compared in Fig. 2E, the y axis numbers do not add up. Presumably at least one of the subsets is present at 10x higher numbers than the y axes indicate, if the different populations are to add up to the total numbers of CD4 or CD8 cells given in the left panels of Fig. 2E. If the mis-calibrated one is the naïve T cells, then the panels in Fig. S2C are probably mislabeled.

While assembling Figure 2C, some panels were incorrectly inserted and mislabeled. This has now been corrected, and the new figure in the revised manuscript includes panels with correctly labelled frequency and cell number information. In addition, we checked all other figures and confirmed that similar errors had not been introduced in the figures for the other analyzed tissues. We thank the reviewer for noting these issues, and we apologize for the errors that were made.

The reviewer also requested that we show actual flow cytometry plots for the T cell analyses. Comment #5 in the reviewer's "Minor Points" also requested the inclusion of standard flow cytometry plots that compared wild type and mutant cells. The revised manuscript contains new FACS plots showing representative stainings for plasma cells/plasmablasts, memory B cells (Supplemental Figure S3) as well as mature T cell subpopulations (Supplemental Figure S4) for young and middle aged WT and URE Δ/Δ mice.

Minor points:

1. *There have been several reports from the Rudd group about fetal and early postnatal CD8 T cells labeled by a time-stamping method. These have identified early-generated cells which also persist long into adulthood and have distinctive effector function profiles. In principle, those authors should be looking at some of the same cells as Montecino-Rodriguez et al. Are the phenotypes of the delta URE homozygous CD8⁺ T cells consistent?*

The fetal CD8 T cells described by the Rudd group were characterized by their CD44^{high} CD122^{high} phenotype in addition to upregulation of multiple migration, activation and natural killer antigens as well as expression of various transcription factors. Since CD44 labeling was included in our immunostainings, we were able to confirm after re-analysis that CD8⁺ CD44^{high} T cells were present in the large intestine of young URE Δ/Δ mice, indicating that it is likely that the CD8 T cells described by Rudd and colleagues were present in URE Δ/Δ mice. This point is noted in the revised Results. However, a definitive conclusion that the CD8 T cells described by the Rudd group are present in URE Δ/Δ mice would be dependent on more extensive phenotypic and gene expression studies. Since this issue is not central to our observations, we hope that the reviewer will agree that such experiments are beyond the scope of the current manuscript.

2. *The altered subset distributions of the mutant PU.1-low cells are interpreted as being affected mostly by the types of progenitors they came from. But is it certain that the reduced levels of PU.1 in the delta URE homozygous B cells are not continuing to make them functionally abnormal as mature cells?*

Our paper presented no evidence that B cells in URE Δ/Δ mice were functionally abnormal and suggested that they had significant functional potential. That conclusion is consistent with two papers from the Nutt group (*Nat Comm* 8:1426, 2017. *Blood* 106:2083, 2005) showing that the number, subset makeup, and function of B cells

was normal in mice with conditional deletion of PU.1 in mature B cells. The reason for this is that there is redundancy between PU.1 and SpiB. B cell function is compromised only when both PU.1 and SpiB are inactivated and/or altered. The revised Discussion cites and briefly discusses these papers.

3. *Reference Vergani et al., 2022, distinguishes between fetal and early postnatal precursors, but still agrees with Montecino-Rodriguez et al. that all these precursors are very different from later lymphoid precursor populations in fully mature adult mouse bone marrow. Here, the authors strongly argue that the delta URE model shows that the relevant precursors have to be fetal. But is there strong evidence that the real cutoff of developmental capability is exactly at birth for delta URE precursors, or whether it could occur some time slightly later? That is a slightly different question than the one the authors previously addressed when they showed the ontogeny of naturally PU.1-low cells in wildtype mice.*

As noted above, we never proposed that there is an absolute cutoff of fetal precursors in URE Δ/Δ mice at birth. In fact, the data in Supplemental figures 1 and 2 demonstrate that fetal stem and progenitor cells can produce B and T cell progeny into young adulthood. As noted, this is consistent with previous studies from the Eaves group (*Proc Natl Acad Sci* 104:5878, 2007) and a paper from our laboratory showing that fetal derived B-1 progenitors are maintained in adult bone marrow for several weeks after birth (*Nat Immunol* 7:293, 2006). The fact that fetal progenitors disappear from the marrow by middle age is likely due to their limited self-renewal potential and ultimate exhaustion due to differentiation.

4. *Could the authors capitalize the acronyms "SI" and "LI" for small intestine and large intestine, in the text and figures?*

This change has been made throughout the text and figures.

5. *Use of UMAP dimensional reductions for multiparameter flow cytometry is clever and may inspire other authors to use a similar display. However, there are recognized clues to normal or abnormal status of cells based on particular pattern distortions of the flow biplots in certain pathological situations, and the UMAP plots obscure these signs. It is a shame that the authors did not include more typical displays of flow cytometry data comparing the wildtype and mutant cells: most of it is in Fig. S2, which only shows wildtype data. Additional supplementary figures with more examples comparing wildtype and mutant phenotypes at the same time point would be helpful.*

We thank the reviewer for recognizing the usefulness of using the UMAP dimensional reduction strategy we used to display our data. We analyzed multiple B and T cell subpopulations in multiple tissues from multiple mice of different ages. This resulted in a massive amount of data that was difficult to present and compare using conventional flow plots, which is why we elected to use UMAP presentations.

As noted in the response to comment 2 above, the reviewer requested that actual flow cytometry plots for the B and T cell analyses be shown. Supplemental Figure 3 presents representative FACS plots of different B cell populations and Supplemental Figure 4 shows various T cell subsets in different aged WT and URE Δ/Δ mice.

The reviewer correctly notes that UMAP representation can obscure individual pattern distortions, and it is also very susceptible to noise and outliers in the data sample. During our analyses, we used standard flow best practices to minimize variations between samples during acquisition: all samples were run on the same LSRII instrument with identical parameter settings to control for fluorescence intensity variation between runs. We also followed the recommended best practice for UMAP analyses: identical number of acquired cells from WT and URE Δ/Δ samples were concatenated by age and tissue before running the UMAP algorithm (*FlowJo10*). In this case, flow samples that included events that were too low or out of average range of fluorescence intensities for the parameters analyzed were excluded. These points are now stated in the revised methods section.

Reviewer #2

1. *To illustrate that adult lymphopoiesis is strongly reduced, numbers of T and B cells in the spleen and thymus are low with few naïve lymphocytes being found whereas activated memory cells are at levels close to normal. This finding suggests indeed adult lymphopoiesis is reduced although residual production may occur because FigS1 shows the presence at very low levels of B220+IgM- cells in adult BM and the thymus as mentioned above is active until 7 weeks of age at least. So there is no formal evidence that the activated peripheral cells originate from embryonic and perinatal lymphocytes rather than from low numbers of lymphocytes produced throughout life. Evaluating the presence of N sequences in the V-D-J rearrangements of peripheral B and T cells might contribute to clarify this issue since embryonic lymphocytes exhibit poor repertoire diversity.*

As discussed in the response to Reviewer 1, we detected a low number of B and T cell progenitors in young adult URE Δ/Δ mice. As noted, this can be attributed to the fact that there is not an abrupt loss of fetal derived stem/progenitor cells at birth. Instead, these transient populations persist through young adulthood and gradually decline in number over time. As shown in the new Supplemental Figures 1 and 2, the expanded phenotypic analysis of lymphopoiesis in the bone marrow and thymus demonstrated that PU.1^{low} fetal lymphoid progenitors are not sustained long-term and B and T cell development had completely ceased in URE Δ/Δ mice by middle age. Further evidence that this had occurred is demonstrated by the data showing that Peyer's patches and Mesenteric Lymph nodes, which require naïve B and T cells to be maintained, had completely disappeared in URE Δ/Δ mice.

2. *Another potential limitation is the absence of adult competitors in the experimental setting. It is not evident that in the presence of diverse population of B and T cells the lymphocytes originated perinatally would persist and contribute to adult immunity. A competitive reconstitution experiment might provide additional evidence that these embryonic lymphocytes persist in normal animals.*

As noted in the response to Reviewer 1, experiments involving transplantation of PU.1 deficient stem and progenitor cells would not be informative, as such cells do not function well in transplantation models. We considered transplanting WT stem/progenitor cells from adult CD45.1 mice into young CD45.2 URE Δ/Δ mice to test if the WT cells outcompeted the URE Δ/Δ fetal derived cells in tissues over time. However, the drugs or irradiation needed to deplete resident stem/progenitors from the bone marrow to allow engraftment could compromise the fetal derived, mature B and T cells resident in tissues such as the large intestine. We thus decided to not use transplantation approaches because they are noisy. We nevertheless tried to address the issue raised by the reviewer.

First, the revised results note that if the B and T cell populations present in adult URE Δ/Δ mice were maintained long-term due to the absence of adult lymphopoiesis, they should be widely distributed in multiple secondary lymphoid tissues for extended periods of time. New data in the revised manuscript show that, even though plasma cells were present in the spleen and intestines, they were not detected in the bone marrow. That result is included as a new Figure 6I.

We also compared our results to published data from Vergani et al. (*Immunity* 55:1829, 2022) in which B cells derived from fetal/neonatal progenitors were time stamped on day 19 after birth. Their results showed that IgA⁺ cells derived from fetal/neonatal progenitors were present in the small intestine through at least 18 months of age even though adult lymphopoiesis was intact in the animals that were analyzed.

Vergani et al. further demonstrated that there was a ~40-50% decline in the frequency of marked IgA⁺ plasma cells in the small intestine by 12 months of age. This is exactly what we observed in the small intestine of URE Δ/Δ mice. For example, there were on average ~122,000 IgA⁺ cells in the small intestine of 4-20 and ~60,000 IgA⁺ cells in the small intestine of 22-46 week old URE Δ/Δ mice, which represents a 51% loss. Thus, our data are in remarkable agreement with the results from Vergani et al.

Finally, it is important to note that Vergani et al. could not determine if the observed decline in IgA⁺ cells occurred because of limited life span or because there was competition from B cells generated during adult lymphopoiesis. Since there is no competition from adult B cells in URE Δ/Δ mice, our results indicate that the

gradual loss of IgA⁺ cells in the small intestine is mostly due to their life span. The limitations of the URE Δ/Δ mouse model and the comparisons with the data from Vergani et al. are included in the second to the last paragraph in the revised Discussion, which points out that a full understanding of the role played by fetal derived B cells emerges when the results from lineage tracing studies and URE Δ/Δ mice are integrated.

3. The claim that the activated cells of embryonic origin contribute to tissue homeostasis is overinterpreted and could simply result from heterogeneity in the samples. The differences at week 27-28 is less impressive than between WT weeks 4 to 27. Can the authors put some quantification in this analysis? Otherwise, I would recommend turning down this claim.

We do not understand the point raised by the reviewer regarding heterogeneity in the samples, but in an attempt to address it, we have revised various sections of the manuscript to temper the conclusion that activated cells of embryonic origin contribute to tissue homeostasis.

4. It has been previously shown that in the absence of thymic input thymopoiesis can persist for long periods of time (Peaudecerf et al J Exp Med 2012; Martins et al J Exp Med 2012, Nature 2014). The authors might like to discuss their data in this context.

The revised discussion (paragraph 2) cites the two JEM papers and discusses them in the context of the new data in Supplemental Figure 2 showing that there is not persistent thymopoiesis in URE Δ/Δ mice.

5. Another confusing element is that in the schematic representation of hematopoiesis the embryonic/perinatal lymphocyte production is HSC independent although in Montecino-Rodriguez et al 2016 it is stated that the HSC independent wave of yolk sac origin that contributes to the first B1 cells is missing in these mutants. This requires clarification.

The reviewer correctly notes the yolk sac wave of B-1 development is missing from URE Δ/Δ mice, as it does not occur when *Spi1* expression is 20% of normal. However, our 2016 *Immunity* paper showed that other fetal waves of B-1 lymphopoiesis were intact, and the results of the current paper show that these latter waves are sufficient to fill the B-1 B cell compartment in the peritoneal cavity. Paragraph 7 of the revised Discussion makes this point.

In summary, the data in our paper clearly document that transient PU.1^{low} fetal progenitors generate T and B cells, and B-1 and MZ B cells in particular, that colonize multiple secondary lymphoid tissues of young URE Δ/Δ mice. The new supplemental data showing that B and T cell development is blocked in adult URE Δ/Δ mice makes the additional case that the activated and memory B and T cells found in various secondary lymphoid tissues, and the intestines in particular, through middle age are the progeny of those transient PU.1^{low} fetal progenitors.

We again thank the reviewers for constructive comments, and we hope the manuscript is now acceptable for publications.

Sincerely,

Kenneth Dorshkind, Ph.D.

May 22, 2024

RE: Life Science Alliance Manuscript #LSA-2024-02629-TR

Dr. Kenneth Dorshkind
David Geffen School of Medicine at UCLA
Department of Pathology and Laboratory Medicine
19833 Le Conte Avenue
Los Angeles, California 90095

Dear Dr. Dorshkind,

Thank you for submitting your revised manuscript entitled "Transient PU.1 Low Fetal Progenitors Generate Lymphoid Progeny that Contribute to Adult Immunity". We would be happy to publish your paper in Life Science Alliance pending final revisions necessary to meet our formatting guidelines.

- please be sure that the authorship listing and order is correct
- please add the Twitter handle of your host institute/organization as well as your own or/and one of the authors in our system
- please move your main, supplementary figure, and table legends to the main manuscript text after the references section
- please remove legends from the supplementary figures. The legends should appear only in the manuscript file.
- please add callouts for Figures S3A-C and S4A-B to your main manuscript text

A. FINAL FILES:

B. MANUSCRIPT ORGANIZATION AND FORMATTING:

**Submission of a paper that does not conform to Life Science Alliance guidelines will delay the acceptance of your

manuscript.**

The license to publish form must be signed before your manuscript can be sent to production. A link to the electronic license to publish form will be available to the corresponding author only. Please take a moment to check your funder requirements.

Sincerely,

May 23, 2024

RE: Life Science Alliance Manuscript #LSA-2024-02629-TRR

Dr. Kenneth Dorshkind
David Geffen School of Medicine at UCLA
Department of Pathology and Laboratory Medicine
Pathology and Laboratory Medicine
19833 Le Conte Avenue
Los Angeles, California 90095

Dear Dr. Dorshkind,

Thank you for submitting your Research Article entitled "Transient PU.1 Low Fetal Progenitors Generate Lymphoid Progeny that Contribute to Adult Immunity". It is a pleasure to let you know that your manuscript is now accepted for publication in Life Science Alliance. Congratulations on this interesting work.

DISTRIBUTION OF MATERIALS:

Again, congratulations on a very nice paper. I hope you found the review process to be constructive and are pleased with how the manuscript was handled editorially. We look forward to future exciting submissions from your lab.

Sincerely,
